# A structural basis for the diverse linkage specificities within the ZUFSP deubiquitinase family

Thomas Hermanns [1], Christian Pichlo [2], Ulrich Baumann [2] & Kay Hofmann [1✉]

Eukaryotic deubiquitinases are important regulators of ubiquitin signaling and can be sub-divided into several structurally distinct classes. The ZUFSP family, with ZUP1 as its sole human member, has a modular architecture with a core catalytic domain highly active against the ubiquitin-derived peptide RLRGG, but not against ubiquitin itself. Ubiquitin recognition is conferred by additional non-catalytic domains, making full-length ZUP1 active against long K63-linked chains. However, non-mammalian ZUFSP family members contain different ubiquitin-binding domains in their N-terminal regions, despite their high conservation within the catalytic domain. Here, by working with representative ZUFSP family members from insects, fungi and plants, we show that different N-terminal domains are associated with different linkage preferences. Biochemical and structural studies suggest that the acquisition of two family-specific proximal domains have changed the default K48 preference of the ZUFSP family to the K63 preference observed in ZUP1 and its insect homolog. Additional N-terminal zinc finger domains promote chain cleavage without changing linkage-specificity.

---

[1] Institute for Genetics, University of Cologne, Zülpicher Straße 47a, 50674 Cologne, Germany. [2] Institute of Biochemistry, University of Cologne, Zülpicher Straße 47, 50674 Cologne, Germany. ✉email: kay.hofmann@uni-koeln.de

Posttranslational modifications of proteins by the covalent attachment of ubiquitin regulate a plethora of cellular pathways, including proteostasis, vesicular trafficking, inflammation, and the DNA-damage response[1]. The ability of ubiquitin to form chains of various linkage types adds to the complexity of the ubiquitin system[2]. Deubiquitinating enzymes (DUBs) are essential components of ubiquitin-based signal transduction: they are required for the maturation of ubiquitin precursor proteins and also cleave ubiquitin chains and remove ubiquitin from substrate proteins, thereby reversing the action of ubiquitin ligases and erasing the ubiquitin signal. Different types of ubiquitin chains convey different messages and many DUBs discriminate between linkage types by showing a more or less pronounced cleavage selectivity[3]. The human genome encodes more than 100 DUBs, belonging to seven different classes as defined by their structural features and evolutionary ancestry[4].

The seventh and most recently discovered deubiquitinase class is the ZUFSP family, with ZUP1 as its sole human member[5–8]. Human ZUP1 was found to be highly specific for K63-linked chains, with a strong preference for long chains and little activity against di-ubiquitin. The crystal structure of ZUP1 in a covalently linked complex with mono-ubiquitin revealed a modular architecture, with a catalytic core domain highly active against the ubiquitin-derived peptide RLRGG, but unable to cleave ubiquitin chains[5]. The recognition of ubiquitin is mediated by a number of ubiquitin-binding domains, including a novel type (zUBD) used for positioning the outgoing (S1) ubiquitin and a MIU domain in a position suitable to bind the upstream (S2) ubiquitin in a K63 linkage[5]. The structure also contains a helix-turn-helix motif (α2/3), positioned at the expected binding site (S1′) for the proximal ubiquitin. Since only the S1 ubiquitin was part of the structure, the role of the MIU and the α2/3 region can only be estimated. The N-terminal part of the human ZUP1 protein contains several Zinc fingers resembling the ubiquitin-binding (UBZ) consensus[9]; these domains are not part of the published ZUP1 structures, but their expected position is compatible with the recognition of ubiquitin units further distal of the cleavage site[5].

Like mammals, the fission yeast *Schizosaccharomyces pombe* encodes a single ZUFSP family member, Mug105, which is much shorter than human ZUP1 and lacks all recognizable ubiquitin-binding domains. Interestingly, this fungal protein preferentially cleaves K48-linked ubiquitin chains. Since human ZUP1 and *S. pombe* Mug105 are bona fide orthologs, an analogous function of these enzymes can be expected[10]. It is therefore highly surprising that two orthologous DUBs have evolved fundamentally different linkage specificities. A bioinformatical survey of ZUFSP family members from different taxa had revealed a well-conserved catalytic domain, but a great diversity in associated ubiquitin-binding domains (UBDs)[5,7]. This diversity raises the question if (and how) these UBDs determine the cleavage specificity of the ZUFSP family. To address this question, we selected five different members of the ZUFSP family with different UBDs, assayed their activity and linkage specificity, solved the crystal structure for two of them, and addressed the linkage specificity by testing several structure-guided mutations. Our results suggest that the K63 linkage specificity of metazoan ZUP1 members is caused by the interplay of the zUBD domain with the α2/3 helices, while the more distal UBDs mostly contribute to avidity and the long-chain preference. We also show that human ZUP1 works at markedly sub-optimal speed due to an unfavorable oxyanion hole, and can be made considerably more active by a single point mutation gleaned from the insect ZUP1 structure.

## Results

**Structure of Mug105**. In our initial biochemical analysis of Mug105, the sole ZUFSP family member from *S. pombe*, we had found striking differences in cleavage specificity compared to its human counterpart: unlike ZUP1, the fungal enzyme selectively cleaved K48 chains and did not show a preference for very long ubiquitin chains[5]. Since sequence analysis of Mug105 did not identify any ubiquitin-binding domains, we set out to investigate the basis of Mug105 specificity by solving the crystal structure of full-length Mug105 to a resolution of 2.0 Å (Table 1). The asymmetric unit contained two Mug105 molecules. Except for two short linker regions (149–153 and 189–193), the complete Mug105 structure could be built from the electronic density by molecular replacement based on the catalytic domain of human ZUP1 (PDB:6EI1). As shown in Fig. 1a, Mug105 assumes a globular α/β papain-fold structure. The active site residues (Cys-42, His-165, and Asp-183) assume a geometry similar to other papain-fold cysteine proteases (Fig. 1b, Supplementary Fig. 1a). Overall, the Mug105 structure bears a striking resemblance to the ZUP1 catalytic core domain, with an RMSD of 1.7 Å over 215 residues (Fig. 1c). As predicted by sequence analysis, all ubiquitin-binding domains of ZUP1 are absent from Mug105, and no additional UBD candidates are apparent from the structure. The Mug105 structure does not contain ubiquitin, but the recognition site of the ubiquitin C-terminal R-x-R motif appears to be conserved between ZUP1 and Mug105. As shown in Fig. 1d, Asp-89 of Mug105 corresponds to Asp-406 of ZUP1, which forms a salt bridge with Arg-72 of ubiquitin. Likewise, Glu-109 of Mug105 corresponds to Glu-428 of ZUP1, forming a salt bridge with Arg-74 of ubiquitin (Fig. 1d). Similar to what has been observed for ZUP1, these Mug105 residues are required for

### Table 1 Data collection and refinement statistics.

| | Mug105 | TcZUP::Ubiquitin-PA |
|---|---|---|
| PDB ID | 7OIY | 7OJE |
| Wavelength (Å) | 0.968 | 1.000 |
| Resolution range | 39.46–2.05 (2.124–2.05) | 51.67–2.05 (2.123–2.05) |
| Space group | P 41 | P 61 |
| Unit cell | 88.24 88.24 111.01 90 90 90 | 151.61 151.61 83.75 90 90 120 |
| Unique reflections | 53,097 (5246) | 67,359 (6328) |
| Multiplicity | 6.0 (6.2) | 8.6 (7.4) |
| Completeness (%) | 99.76 (99.39) | 97.93 (92.88) |
| Mean I/sigma(I) | 10.4 (1.1) | 12.84 (1.1) |
| R-meas | 0.138 (1.65) | 0.150 (1.87) |
| R-pim | 0.056 (0.66) | 0.051 (0.67) |
| CC1/2 | 0.998 (0.441) | 0.998 (0.411) |
| Reflections used in refinement | 53,081 (5246) | 67,342 (6318) |
| Reflections used for R-free | 2000 (195) | 1927 (180) |
| R-work/R-free (%) | 17.26/20.39 | 18.52/21.86 |
| RMS (bonds) | 0.003 | 0.006 |
| RMS (angles) | 0.5 | 0.78 |
| Ramachandran favored (%) | 96.69 | 97.45 |
| Ramachandran allowed (%) | 2.69 | 2.43 |
| Ramachandran outliers (%) | 0.62 | 0.12 |
| Rotamer outliers (%) | 0.45 | 0.26 |
| Clashscore | 2.39 | 1.8 |
| Average B-factor | 45.57 | 61.63 |
| Macromolecules | 45.28 | 62.69 |
| Ligands | 64.8 | 50.88 |
| Solvent | 49.34 | 44.34 |

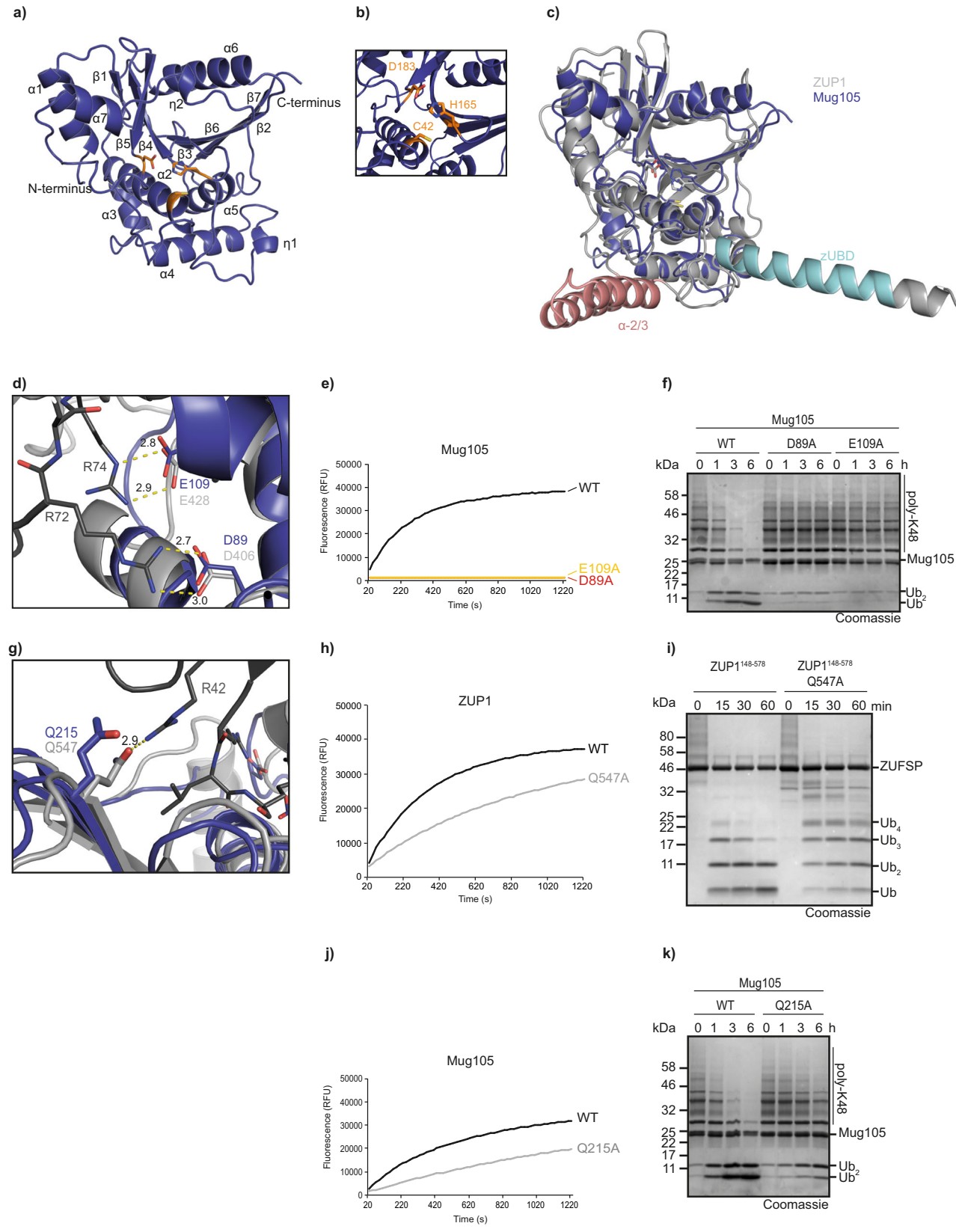

cleavage of the ubiquitin-derived peptide substrate RLRGG-AMC (Fig. 1e) and K48-linked ubiquitin chains (Fig. 1f).

Besides the zUBD region, which interacts with the Ile-44 patch, and the acidic R-x-R interactors, the human ZUP1 structure shows one additional ubiquitin contact within the catalytic core domain: Gln-547 interacts with Arg-42 of ubiquitin (Fig. 1g) and

the Q547A mutant showed a moderate decrease in cleavage activity (Fig. 1h, i). In Mug105, the corresponding residue is Gln-215 (Fig. 1g). The analogous Q215A mutation in Mug105 shows a reduced activity against the RLRGG-AMC peptide (Fig. 1j) and against K48 chains (Fig. 1k). Since the recognition of the ubiquitin-derived peptide should be independent of Arg-42

**Fig. 1 Crystal structure of ZUFSP family member Mug105. a** Overview of the crystal structure in cartoon representation. The catalytic core of Mug105 is shown in blue. The catalytic triad is shown as orange sticks. **b** Magnification of the active site of Mug105 (blue). The catalytic triad is shown as orange sticks. **c** Structural superposition of Mug105 (blue) and ZUP1 (6EI1, gray). The RMS distance is 1.7 Å over 215 residues. The characteristic helical protrusions of ZUP1 are colored cyan (zUBD) and red (α-2/3). **d** Recognition of ubiquitin C terminus by the catalytic core of Mug105. Mug105 (blue) and the ZUP1(light gray)/ubiquitin (dark gray) complex are superimposed and shown in cartoon representation. Key residues are highlighted as sticks. Salt bridges are indicated by dotted lines. **e** Activity of C-terminus recognition mutants (Mug105 D89A or E109A) against RLRGG-AMC. The RFU values shown are the means of triplicates. **f** Activity of mutants described in **e** against K48-linked ubiquitin chains. **g** Recognition of ubiquitin Arg-42. ZUP1 Gln-547 (light gray) forms a hydrogen bond (indicated by yellow dotted line) with ubiquitin Arg-42 (dark gray). The superimposed Gln-215 of Mug105 is conserved and also able to form a hydrogen bond with ubiquitin Arg-42. **h, j** Activity of mutants ZUP1[148–578] Q547A (**h**) or Mug105 Q215A (**j**) against RLRGG-AMC. The RFU values shown are the means of triplicates. **i** Activity of ZUP1[148–578] Q547A against K63-linked ubiquitin chains. **k** Activity of Mug105 Q215A against K48-linked ubiquitin chains. Source data are provided as a Source Data file.

binding, the role of Gln-215 is probably more structural than focused on ubiquitin recognition. Since Mug105 lacks the Ile-44 recognizing zUBD helix and other structural elements that might replace the zUBD, it is unlikely that the Ile-44 patch of ubiquitin is important for recognition by Mug105 (Supplementary Fig. 1b). Accordingly, Mug105 reactivity with the activity-based probe Ub-PA was not impaired by the introduction of an I44A mutation (Supplementary Fig. 1c).

**ZUFSP proteins from other species show different specificities.** The differences in chain selectivity exhibited by human ZUP1 and fungal Mug105 might be caused by the presence of N-terminal UBDs in ZUP1 and their lack in Mug105. To further investigate a connection between UBD-usage and linkage specificities, we selected representative members of the ZUFSP family with different UBD architectures (Fig. 2a). As described previously, ZUFSP family members are found in a wide range of taxa, but are absent from most typical model organisms[5]. While fruit flies lack a ZUFSP protein, the red flour beetle *Tribolium castaneum* encodes a ZUFSP-type DUB in which a 5th Zinc-finger appears to replace the MIU domain. This protein, here referred to as TcZUP (systematic name TC05178), has a recognizable zUBD domain, while the region corresponding to the human α2/3 motif lacks detectable sequence conservation. Most plant genomes encode a single ZUFSP family member. In *Arabidopsis thaliana*, the corresponding protein is At5g24680 (here referred to as AtZUP) and contains a single Zinc finger that resembles the last UBZ-like ZF in human ZUP1. Unlike the human protein, AtZUP lacks the zUBD domain and the α2/3 region. The ascomycete fungus *Aureobasidium pullulans* has a ZUFSP family member (ApZUP) with a particularly unusual arrangement of UBDs (Fig. 2a). Similar to human ZUP1, ApZUP contains a MIU domain preceded by a UBZ-like Zinc-finger. However, ApZUP contains a 2nd Zinc finger after the MIU domain, at a position where human ZUP1 has the zUBD and α2/3 domains.

All three of the described ZUFSP homologs were successfully expressed in the *E. coli* system and tested for DUB activities in a number of different assays. The fungal ApZUP was poorly soluble and found to be inactive against a model substrate and various activity-based probes (Supplementary Fig. 2a, b). This lack of activity might have been caused by an atypical sequence stretch around the catalytic cysteine, which is well-conserved in other ZUFSP family members (Supplementary Fig. 2c). We, therefore, focused our analysis on insect TcZUP and plant AtZUP, which both did show DUB activity in multiple assays. In the RLRGG-AMC cleavage assay, both TcZUP and AtZUP surpassed the already quite impressive activity of human ZUP1 (Fig. 2b, Supplementary Fig. 2d); most other deubiquitinases are hardly able to cleave this peptide substrate. When assaying the AMC release from the DUB model substrate ubiquitin-AMC, AtZUP showed a similar activity to human ZUP1, while TcZUP was considerably more active (Fig. 2c). When incubating AtZUP with

di-ubiquitin species of various linkage types, a modest cleavage activity with a preference for the K48 linkage was observed (Fig. 2d and Supplementary Fig. 2e). When performing this assay with tetra-ubiquitin chains as substrates, the observed cleavage activity was slightly increased but showed the same K48 preference (Fig. 2e). By contrast, when subjecting the insect TcZUP to the same assay, the observed cleavage activity was much higher and showed a preference for K63 linkages (Fig. 2f, g and Supplementary Fig. 2e): While AtZUP only partially cleaved the preferred K48-linked di-ubiquitin after 6 h and tetra-ubiquitin after 2 h, TcZUP showed nearly complete conversion of K63 di-ubiquitin after 30 min and complete cleavage of tetra-ubiquitin after only 5 min. In summary, the plant AtZUP is about as active as human ZUP1 but prefers K48 over K63 linkages, while the insect TcZUP shares with human ZUP1 the preference for K63 linkages but is considerably more active (Fig. 2h and Supplementary Fig. 2f, g). Like human ZUP1[5], Mug105, TcZUP, and AtZUP reacted readily with the ubiquitin activity-based probe Ub-PA; less reactivity was observed for the ubiquitin-like probe NEDD8-PA, while SUMO2-PA and LC3B-PA did not react at all, even after 6 h of incubation (Supplementary Fig. 2h, i).

**Ubiquitin recognition by insect TcZUP.** To further investigate the increased activity of TcZUP and the basis for its K63 linkage selectivity, we reacted a catalytically active fragment TcZUP[218–592] (excluding the first four ZF domains) with the activity-based probe Ubiquitin-PA and solved the crystal structure of the covalent adduct to a resolution of 2.0 Å (Table 1). The asymmetric unit contained two TcZUP:Ub complexes. Except for the first five residues and the linker between the UBZ and the zUBD domains (residues 258–262), all residues of the TcZUP[218–592] fragment could be built into the electronic density. As shown in Fig. 3a, the catalytic core domain of TcZUP[218–592] assumes a globular papain-like fold with a catalytic triad consisting of Cys-371, His-505 and Asp-526 (Fig. 3b and Supplementary Fig. 3a). Similar to human ZUP1, the insect protein has two helical protrusions: α1, which corresponds to the zUBD domain and contacts the Ile-44 patch of the bound distal (S1) ubiquitin, and the α2/3 helix-turn-helix region, which is positioned on the opposite side of the catalytic center and might interact with the proximal (S1′) ubiquitin, which is not present in the structure. The arrangement of zUBD, α2/3, and catalytic core is very similar to the human ZUP1 structure (Fig. 3c), with a RMSD of 1.3 Å over 304 residues. A major difference between the human and insect structures is the presence of an N-terminal MIU domain in ZUP1, which is replaced by an UBZ-like zinc finger in TcZUP (Fig. 3c). Both MIU and UBZ are established types of ubiquitin-binding domains, and their positioning in the structure suggests a role in binding a more distal (S2) ubiquitin unit within the cleaved chain. Both the ZUP1 and TcZUP structures contain only the zUBD-bound S1 ubiquitin, leaving the ubiquitin-binding sites of the MIU and UBZ domain unoccupied.

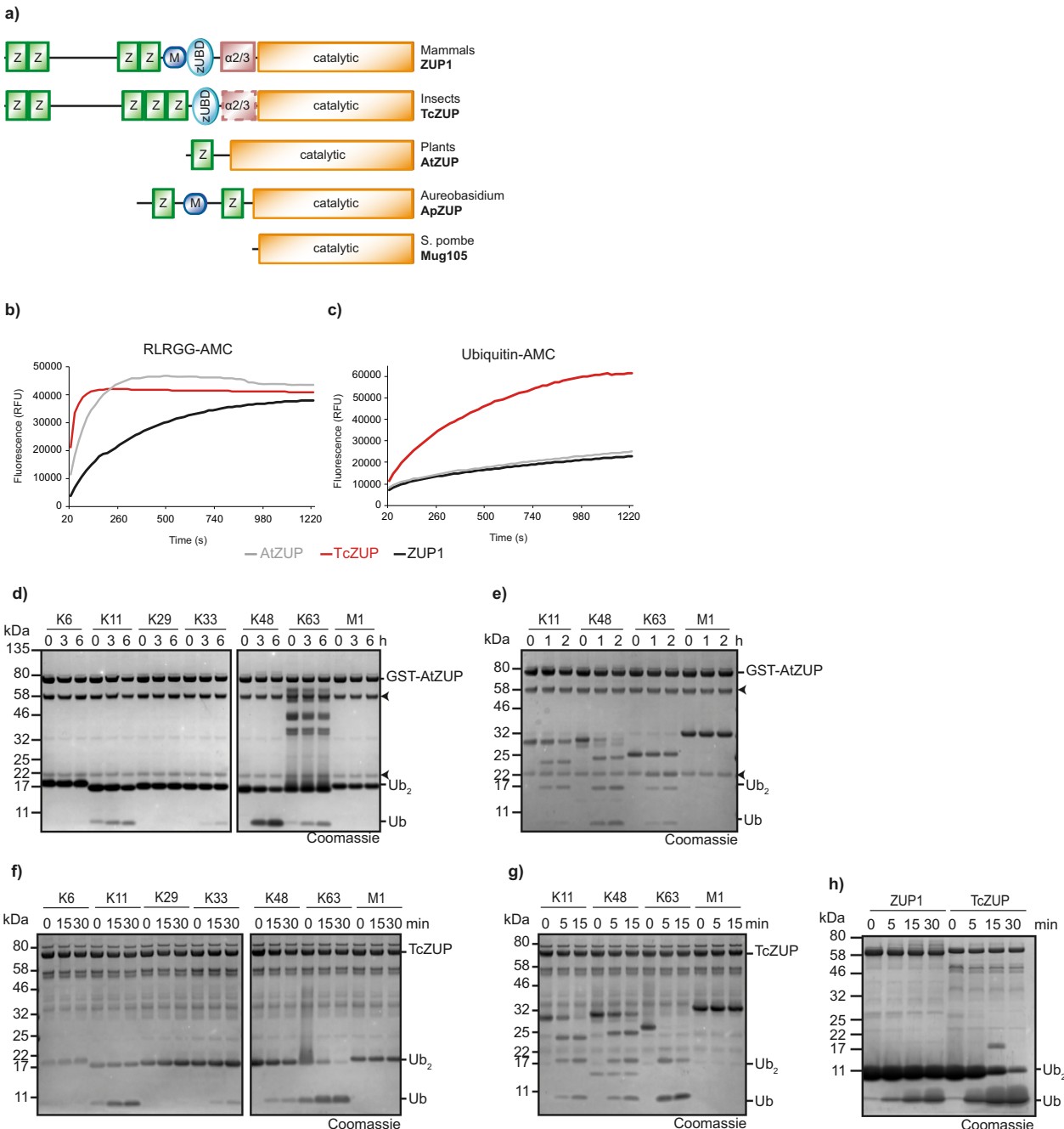

**Fig. 2 Selectivity within the ZUFSP family. a** Schematic representation of the domain architecture of ZUP1 homologs from different species. UBZ-like zinc fingers (Z), MIU domains (M), ZUFSP ubiquitin-binding domains (zUBD), α-2/3 region, and the catalytic domain are shown as boxes. The position of the non-conserved α-2/3 region in insect ZUP is indicated by a dashed line. **b**, **c** Activity of ZUP homologs from *A. thaliana* (AtZUP) and *T. castaneum* (TcZUP) against RLRGG-AMC (**b**) and ubiquitin-AMC (**c**). The RFU values shown are the means of triplicates. **d**, **e** Linkage specificity analysis of AtZUP. A panel of di-ubiquitin (**d**) or tetra-ubiquitin (**e**) chains was treated with GST-AtZUP for the indicated time points. Black arrowheads mark the N- and C-terminal fragments of AtZUP (Supplementary Fig. 5c). **f**, **g** Linkage specificity analysis of TcZUP. A panel of di-ubiquitin (**f**) or tetra-ubiquitin (**g**) chains was treated with TcZUP for the indicated time points. **h** Comparison of ZUP1 and TcZUP hydrolysis rate. K63-linked di-ubiquitin was treated with ZUP1 or TcZUP for the indicated time points. Source data are provided as a Source Data file.

While in the ZUP1 structure the MIU domain is rigidly connected to the zUBD by being part of an uninterrupted α-helix[5], the UBZ of TcZUP is connected to the zUBD by a flexible linker (Fig. 3a, c). In the crystal structure of TcZUP, the UBZ folds back and contacts the S1-ubiquitin moiety, which is already bound by the zUBD domain (Fig. 3a). However, this contact does not engage the canonical ubiquitin-interaction surface of UBZ domains[9], suggesting that it might be non-physiological and possibly caused by the

crystal lattice. This idea is supported by experiments using a double mutation of Glu-248 and Ser-249, two residues involved in the unconventional ubiquitin contact of the UBZ domain (Supplementary Fig. 3b). As shown in Supplementary Fig. 3c, the chain-cleaving activity of the E248A/S249A mutant of the catalytic TcZUP fragment is indistinguishable from that of the wild-type form.

The zUBD surface that recognizes the Ile-44 patch of the outgoing (S1) ubiquitin is highly conserved between ZUP1 and

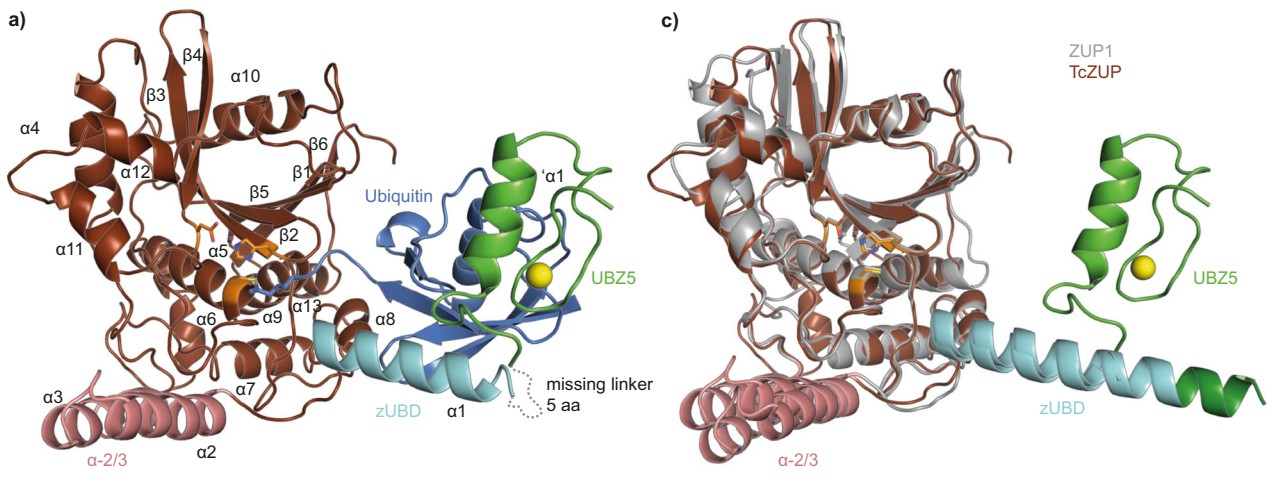

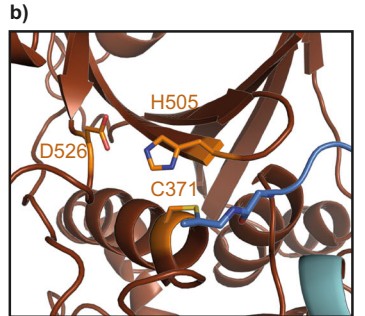

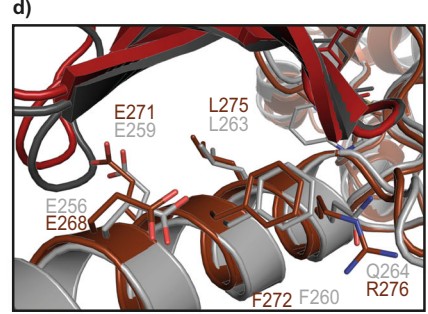

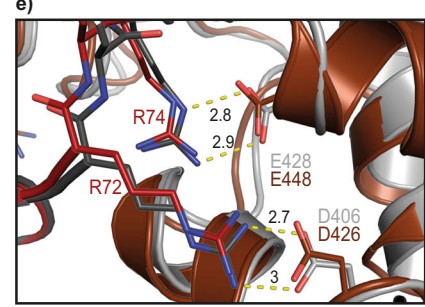

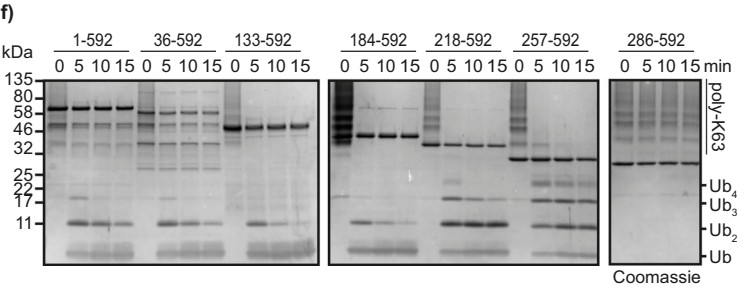

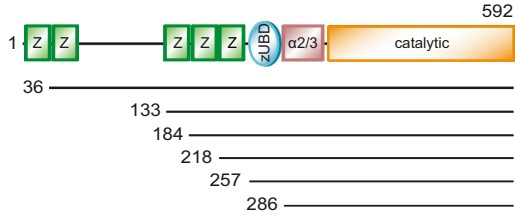

**Fig. 3 Crystal structure of *T. castaneum* ZUP bound to a covalent Ub probe. a** Overview of the crystal structure in cartoon representation. The catalytic core of TcZUP is shown in brown, ubiquitin in blue. The ZnF is colored green, the zUBD region in cyan and the α-2/3 region in red. The catalytic triad is shown as orange sticks. The Zn atom of the ZnF is shown as a yellow sphere. **b** Magnification of the active site of TcZUP (brown). The catalytic triad is shown as sticks and colored orange. **c** Structural superposition of TcZUP (brown) and ZUP1 (6EI1, gray). RMS distance is 1.3 Å over 304 residues. The characteristic helical protrusions of ZUP1 and TcZUP are colored cyan (zUBD) and red (α-2/3). The ZnF of TcZUP and the MIU of ZUP1 are colored light and dark green respectively. **d** Ubiquitin binding by the zUBD is conserved in TcZUP. The zUBD of ZUP1 (light gray) and TcZUP (brown) were superimposed and shown in cartoon representation. Key residues are highlighted as sticks. The corresponding ubiquitins are shown in carton representation and colored dark gray and red, respectively. **e** Recognition of ubiquitin C terminus by the catalytic core of TcZUP. The TcZUP/ubiquitin (brown/red) and the ZUP1/ubiquitin (light gray/dark gray) complexes are superimposed and shown in cartoon representation. Key residues are highlighted as sticks. Salt bridges are indicated by dotted lines. **f** Activity of TcZUP FL and truncations lacking the UBDs against K63-linked $Ub_{6+}$ chains. A mixture of K63-linked poly-ubiquitin chains ($Ub_{6+}$) are cleaved over time to mono-, di-, tri-, and tetra-ubiquitin. The reaction was stopped after the indicated time points, separated by SDS-PAGE and coomassie stained. Source data are provided as a Source Data file.

TcZUP. As shown in Fig. 3d, the contact residues of ZUP1 (Glu-256, Glu-259, Phe-260, Leu-263, and Gln-264) are structurally equivalent to the TcZUP contact residues Glu-268, Glu-271, Phe-272, Leu-275, and Arg-276. The same is true for the recognition of the ubiquitin C terminus: Arg-72 and Arg-74 of ubiquitin form salt bridges with Asp-406 and Glu-428 of human ZUP1, and their corresponding residues Asp-426 and Glu-448 in TcZUP (Fig. 3e). The N-termini of human ZUP1 and insect TcZUP have a similar architecture with multiple UBZ-like zinc fingers, which in ZUP1 are dispensable for catalysis and have a modest influence on the cleavage rate for long ubiquitin chains. To assess the importance of the different TcZUP UBDs, a series of truncations were tested for their activity on K63-linked $Ub_{6+}$ chains. As shown in Fig. 3f, TcZUP truncations lacking the first one, two, or three zinc fingers (TcZUP[36–592], TcZUP[133–592] and TcZUP[184–592]) cleave $Ub_{6+}$ chains with similar kinetics as the full-length form TcZUP1–592.

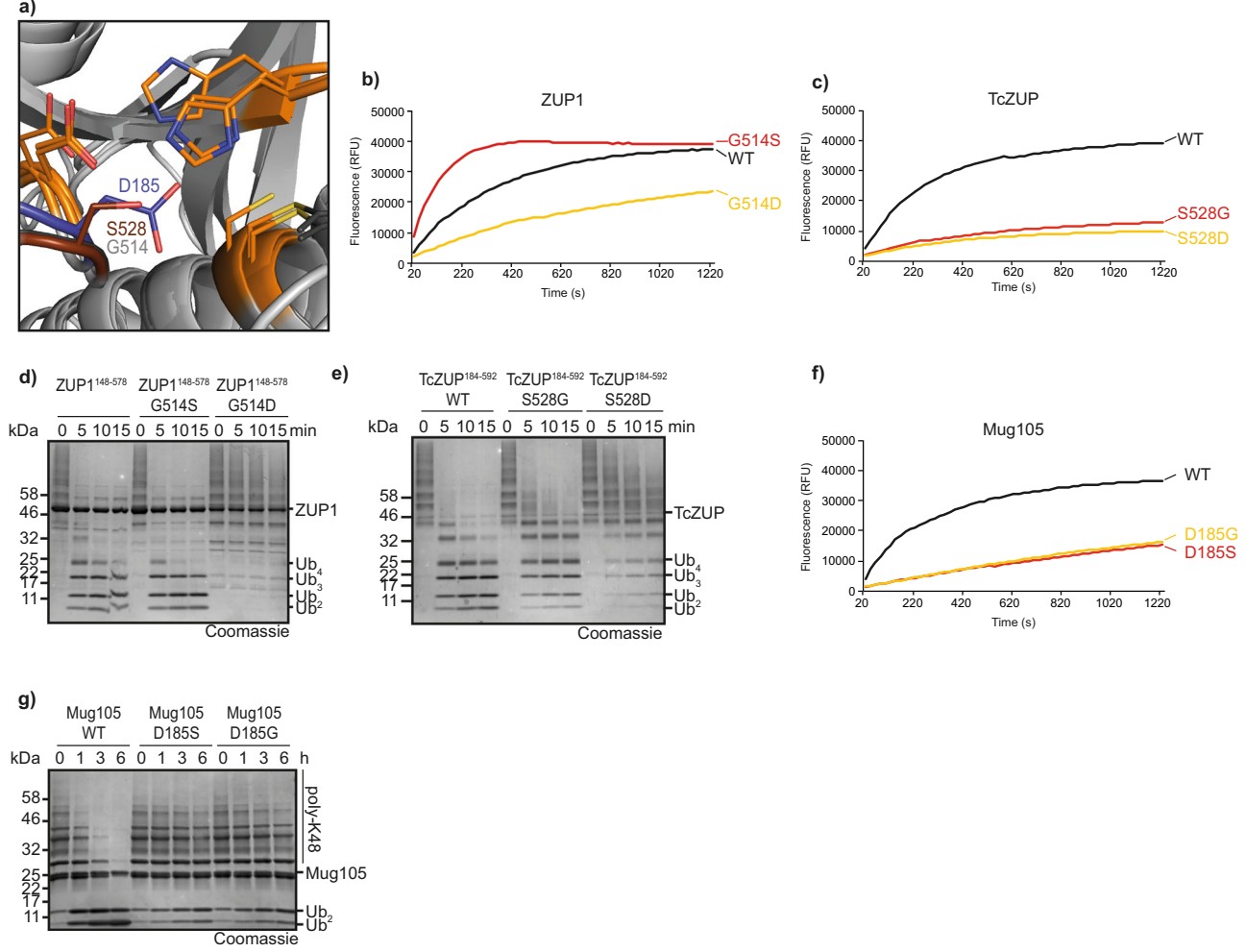

**Fig. 4 Identification and importance of the oxyanion hole. a** Superposition of ZUP1 (PDB:6EI1), TcZUP, and Mug105 active sites. The catalytic core domain is shown in cartoon representation and colored light gray. The catalytic triads are shown as sticks and colored orange. Ser-528 of TcZUP (brown) is in a suitable position to stabilize the tetrahedral reaction intermediate. The corresponding residues Gly-514 (ZUP1) and Asp-185 (Mug105) are shown as sticks and colored light gray and blue, respectively. **b**, **c** Activity of ZUP1[148–578] (**b**) and TcZUP[184–592] (**c**) oxyanion hole mutants against RLRGG-AMC. The RFU values shown are the means of triplicates. **d**, **e** Activity of ZUP1[148–578] (**d**) and TcZUP[184–592] (**e**) oxyanion hole mutants against K63-linked $Ub_{6+}$ chains. **f**, **g** Activity of Mug105 oxyanion hole mutants against RLRGG-AMC (**f**) or K48-linked poly-ubiquitin chains (**g**). The RFU values shown are the means of triplicates. Source data are provided as a Source Data file.

Additional removal of the fourth (TcZUP[218–592]) and fifth (TcZUP[257–592]) zinc finger gradually decreased the chain-cleaving activity, whereas removal of the zUBD (TcZUP[286–592]) rendered the enzyme completely inactive against ubiquitin chains (Fig. 3f and Supplementary Fig. 3d), although it was still able to cleave the peptide substrate RLRGG-AMC (Supplementary Fig. 3e). This gradual loss of chain-cleaving activity was not accompanied by a change in linkage specificity (Supplementary Fig. 3f, g).

**Role of the oxyanion hole**. The hydrolysis rate of cysteine proteases is dependent on the stabilization of the negatively charged tetrahedral transition state, which is usually accomplished by a positively charged pocket, the oxyanion hole[11]. Previously, we had shown that the oxyanion hole of the related proteases UFSP2 and papain is not conserved in human ZUP1, and that a mutation of the corresponding residue Ser-351 in ZUP1 does not impair activity[5]. Since TcZUP is much more active than ZUP1 in all assays, we investigated if this activity difference is due to a better oxyanion hole stabilization by the insect protein. As shown in Fig. 4a, Ser-528 of TcZUP might—despite its unusual position—

help stabilize the tetrahedral reaction intermediate, while the corresponding Gly-514 in human ZUP1 lacks this ability. In order to test the importance of Ser-528, we mutated this residue to glycine (TcZUP[S528G]) and conversely mutated the ZUP1 glycine to serine (ZUP1[G514S]). When testing both mutants for RLRGG-AMC cleavage, ZUP1[G514S] showed a striking activity increase over wild-type ZUP1, while TcZUP[S528G] was considerably less active than wild-type TcZUP (Fig. 4b, c). These results suggest that Ser-528 is partly responsible for the high activity of the insect TcZUP, and that this property can be transferred to the human homolog. To assess if the activity increase of TcZUP and the human mutant ZUP1[G514S] over wild-type ZUP1 is due to a rise in $k_{cat}$ or a decreased $K_m$, we determined the kinetic parameters for RLRGG-AMC cleavage by these three enzymes. As shown in Supplementary Fig. 4a–c, TcZUP has a $k_{cat}$ of 95.6 s$^{-1}$, which is 14.5-fold higher than the $k_{cat}$ of human ZUP1 (6.6 s$^{-1}$), while the $K_m$ values are comparable (89.0 μM vs 91.4 μM). The G514S mutation of ZUP1 increases the $k_{cat}$ by a factor of 3.4 (22.6 s$^{-1}$) while the $K_m$ is only reduced by 30% (69.9 μM). These data suggest that—at least for RLRGG-AMC cleavage—the increased activity of the ZUP1[G514S] is mainly

determined by $k_{cat}$, consistent with a potential role in oxyanion hole stabilization.

The kinetic differences were less pronounced when subjecting the same mutants to ubiquitin chain disassembly assays: The ZUP1[G514S] mutant is more active than the wildtype in degrading K63-linked Ub[6+] chains, but only by a small margin—best visible in the disappearance of tetra-ubiquitin (Fig. 4d, Supplementary Fig. 4d)—while the ZUP1[G514D] mutant is nearly inactive. Conversely, the TcZUP[S528G] is somewhat less active than the TcZUP wildtype, but still able to cleave long K63 chains within minutes (Fig. 4e). These results suggest that Ser-528 of TcZUP is important for the catalytic activity against small substrates, but less so for processing longer chains in a more physiological situation.

In *S. pombe* Mug105, the corresponding position of Gly-514 and Ser-528 is occupied by Asp-185 (Fig. 4a). Unlike ZUP1, Mug105 did not profit from a D185S mutation (Fig. 4f), and the introduction of an aspartate into the human (ZUP1[G514D]) and insect (TcZUP[S528D]) proteins lead to a reduction in catalytic activities in both enzymes (Fig. 4b, c). As expected from the RLRGG-AMC results, both Asp-185 mutants of Mug105 are hardly active against ubiquitin chains (Fig. 4g) suggesting that the beneficial effects of a serine substitution at a position corresponding to Gly-514 in ZUP1 cannot be universally applied to all ZUP1 family members.

**Conformational changes upon ubiquitin binding.** For human ZUP1 and insect TcZUP, structures with covalently bound (S1) ubiquitin are available, while Mug105 could only be crystallized in isolation. When comparing the structures, two highly conserved non-catalytic residues were found to assume different conformations in the Ub-bound and unbound forms. Trp-423 of ZUP1 and Trp-443 in TcZUP are identically positioned next to the ubiquitin C terminus. The corresponding Trp-104 of the ubiquitin-less Mug105 structure is mainly adopting a conformation blocking the substrate-binding cleft (Fig. 5a). A similar conformational change is observed for Gln-489 in ZUP1 and Gln-503 in TcZUP1, which both form a weak interaction with Arg-74 of ubiquitin. In the ubiquitin-less Mug105 structure, the corresponding Gln-163 assumes a different orientation, which would not support a ubiquitin contact (Fig. 5a). We hypothesized that the conformational changes are not due to the species difference, but rather induced upon ubiquitin binding. To investigate the importance of these conformationally flexible residues, we tested their alanine mutations for activity in RLRGG-AMC and chain cleavage assays. For both ZUP1 (Fig. 5b, c) and TcZUP (Fig. 5d, e), the Trp → Ala mutation showed a dramatic loss of activity against the peptide substrate and ubiquitin chains. By contrast, the Gln → Ala mutations hardly showed any effect. In Mug105, the W104A mutation was equally disruptive as in the animal proteins, but Q163A also caused a substantial loss of activity in both assays (Fig. 5f, g).

Unlike other DUB classes, the ZUFSP family lacks the aromatic gatekeeper residue following the catalytic histidine[12]. This role might be filled by the conformationally flexible Trp-423 residue, which contacts the penultimate residue of the outgoing ubiquitin from a different position. To test a possible gatekeeper role of this tryptophan, the reactivity of the Trp → Ala mutant against activity-based probes based on ubiquitin C-terminal variants was assessed. As shown in Fig. 5h, wild-type ZUP1 reacted readily with the Ub-PA probe and also showed some unexpected reactivity with variant ubiquitin probes carrying an alanine (Ub[G75A]-PA) or even valine (Ub[G75V]-PA) at the penultimate position. By contrast, the W423A mutant (ZUP1[W423A]) did not react with the wild-type probe, but maintained reactivity against

the G75A and G75V probes. These data suggest that W423 of ZUP1 is important for recognizing Gly-75 of ubiquitin, but does not prevent access of non-glycine residues at position 75 from accessing the active site.

**Domains involved in AtZUP activity.** Since the plant protease AtZUP has the same K48 preference as fungal Mug105 (Fig. 2d) but contains an additional UBZ-type zinc finger domain not found in the *S. pombe* protein (Fig. 2a), we expected this domain to be dispensable for catalysis and linkage selectivity, but possibly to enhance long-chain degradation—similar to what was observed for the N-terminal UBZ-type regions in the animal ZUFSP proteins. Surprisingly, an N-terminally truncated version (AtZUP[64−399]) was completely inactive against both long K48 chains (Fig. 6a) and di-ubiquitin (Fig. 6b). In the RLRGG-AMC cleavage assay, the UBZ-less form was highly active (Fig. 6c), suggesting that the catalytic center and recognition of the ubiquitin C-terminal R-x-R motif are still intact. Since AtZUP[64−399] reacted readily with the activity-based Ub-PA probe (Fig. 6d), the recognition of the outgoing (S1) ubiquitin appears to be unaffected by the UBZ truncation. A change of linkage selectivity by the truncation is also unlikely, since AtZUP[64−399] does not cleave K63 chains (Supplementary Fig. 5a). Thus, the AtZUP zinc finger appears to have a critical role in linkage-selective chain degradation.

Bacterially expressed AtZUP protein always appeared as a full-length product (running at ~45 kDa) and two smaller products (running at ~28 and ~20 kDa, respectively). Surprisingly, both the full-length 45 kDa band and the 28 kDa band shifted upon incubation with Ub-PA, indicating that the shorter form is catalytically active (Fig. 6d). Intact mass analysis positions the cleavage site after Gly[253]Gly[254], within a poorly conserved and probably unstructured loop specific to plant members of the ZUFSP family (Supplementary Fig. 5b). Since cleaved AtZUP is catalytically active, the 3D structure is not affected by hydrolysis of this particular peptide bond and the two cleavage products still form the proper 3D structure. The internal Gly–Gly cleavage site resembles the C terminus of ubiquitin, suggesting that the fragments might arise from AtZUP autocleavage. However, as shown in Supplementary Fig. 5c, neither mutation of the AtZUP active cysteine (C130A) nor mutation of the cleavage site (G253A/G254A) altered the AtZUP processing. Thus, the observed cleavage might be non-physiological and caused by a bacterial protease targeting unstructured regions. In an attempt to improve crystallization of AtZUP, we generated several internal deletions, replacing the plant-specific insertion around Gly[253]Gly[254] (Supplementary Fig. 5d) by a short glycine-rich linker (GGGGSGGSG). While these deletion mutants did not crystallize, we noted a strong increase in DUB activity over the wild-type version (Fig. 6e, f). The longest internal deletion (AtZUP[Δ219-275]) was able to cleave long K48 chains within minutes, whereas wild-type AtZUP showed little cleavage after 1 h. Increases in activity over the wild-type version were also observed for the other deletions. (Fig. 6e). The greatly increased activity of the deletion mutants was also observed in di-ubiquitin and RLRGG-AMC cleavage assays (Fig. 6f, g). The K48 selectivity was not affected by the substitutions (Fig. 6g).

**Role of the α2/3 region in chain recognition.** In previous work, we had shown that an internal deletion of the ZUP1 α2/3 helix-turn-helix region abrogated chain cleavage, but did not influence the reactivity against Ub-AMC[5]. These data suggest a role for the α2/3 region at the S1′ site coordinating the proximal ubiquitin, which is in agreement with the position of the α2/3 region relative to the active site. The α2/3 helices appear to be rigidly attached to the core domain. Gly-354 is a non-loop residue that needs to be

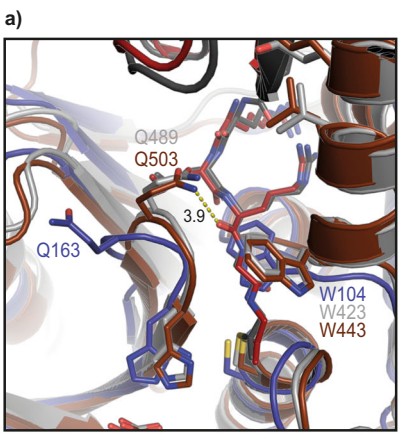

**Fig. 5 Ubiquitin-binding induced conformational changes. a** Structural superposition of ubiquitin-bound TcZUP (brown) and ZUP1 (6EI1, gray) with ubiquitin-less Mug105 (blue). The catalytic core is shown in carton representation and key residues are highlighted as sticks. C-terminal residues of ubiquitin are shown as sticks and colored red and dark gray. Trp-104 and Gln-163 exhibit different conformations compared to the corresponding residues of ZUP1 and TcZUP. **b**, **c** Activity of ZUP1[148-578] W423A or Q489A against RLRGG-AMC (b) or K63-linked $Ub_{6+}$ chains (c). **d**, **e** Activity of TcZUP[187-592] W443A or Q503A against RLRGG-AMC (**d**) or K63-linked $Ub_{6+}$ chains (**e**). **f**, **g** Activity of Mug105 W104A or Q163A against RLRGG-AMC (**f**) or K48-linked ubiquitin chains (**g**). **h** Activity-based probe reaction of wildtype or W423A-mutated ZUP1[148-578] with Ub-PA. Both DUBs were tested against WT, G75A, or G75V Ub-PA. Black arrowheads mark the shifted band after reaction. Source data are provided as a Source Data file.

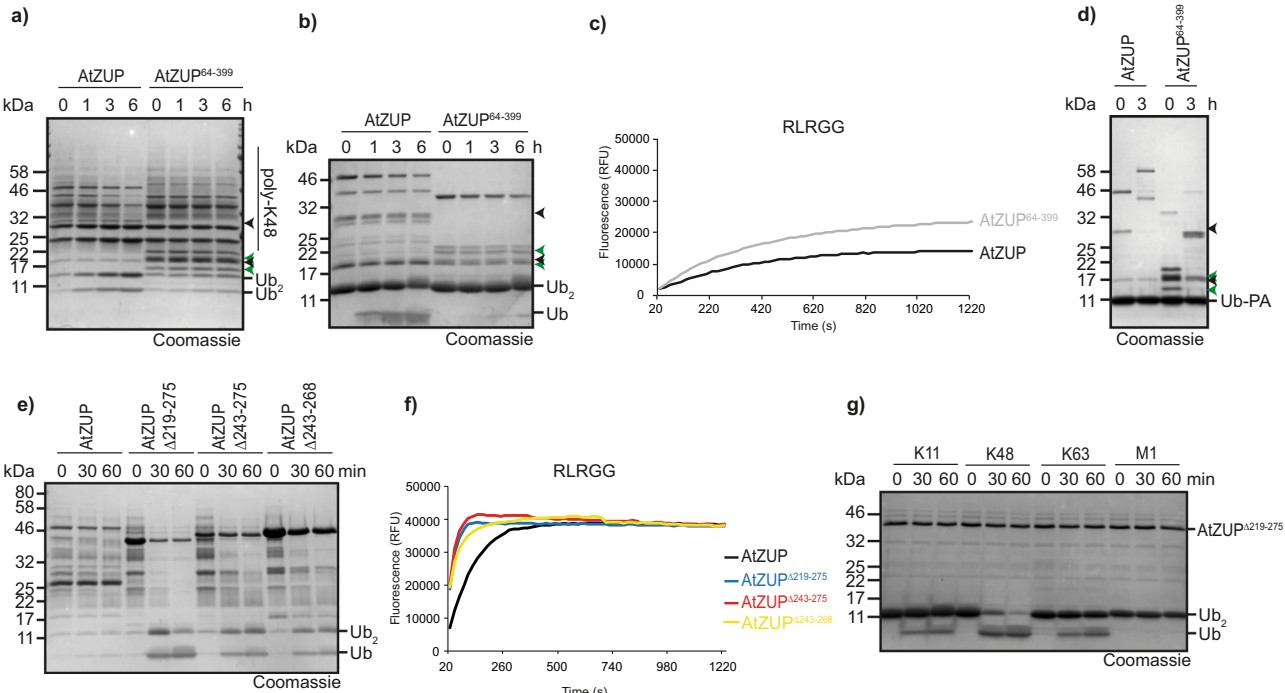

**Fig. 6 Ubiquitin chain cleavage by AtZUP is dependent on UBZ. a–d** Activity assays of FL AtZUP or UBZ-truncated AtZUP[64–399] against K48-linked ubiquitin chains (**a**), K48-linked di-ubiquitin (**b**), RLRGG-AMC (**c**), and Ub-PA (**d**). Arrowheads mark the N- and C-terminal fragments of FL AtZUP (black) or AtZUP[64–399] (green). **e, f** Activity assays of FL AtZUP and the internal truncations AtZUP[Δ219–275], AtZUP[Δ243–275] or AtZUP[Δ243–268] against K48-linked ubiquitin chains (**e**) or RLRGG-AMC (**f**). **g** Linkage specificity analysis with AtZUP[Δ219–275]. A panel of di-ubiquitin chains was treated with AtZUP[Δ219–275] for the indicated time points. Source data are provided as a Source Data file.

small and flexible for allowing proper orientation of the α2/3 loop (Fig. 7a). A disruptive mutation (G354P) at this position phenocopies the deletion of the entire α2/3 loop: both mutants were inactive against ubiquitin chains (Fig. 7b) but remained fully active in the RLRGG-AMC cleavage assay, which is independent of the S1′ ubiquitin recognition (Fig. 7c). The structure of the insect TcZUP also shows an α2/3 loop that superimposes well with the human version (Supplementary Fig. 6a) despite the absence of sequence conservation. This lack of conserved S1′ ubiquitin-binding residues is corroborated by the results of a quintuple mutation in the ZUP1 α2/3 loop: While the deletion of the loop prevented chain cleavage, the simultaneous mutation of five exposed amino acids within the loop (Supplementary Fig. 5a) neither affected chain cleavage nor hydrolysis of RLRGG-AMC (Supplementary Fig. 6b, c).

Since the combination of α2/3 and zUBP is found in K63-cleaving ZUFSP members, but is absent from the K48-recognizing ones, we tested if the linkage preference of the α2/3-less Mug105 can be altered by the addition of a ZUP1-derived N-terminal region (ZnF4-MIU-zUBD-α2/3). However, the chimeric Mug105[ZUP1-NT] construct maintained all cleavage properties of the original Mug105: it was still able to cleave K48 chains (Fig. 7e), did not cleave K63 chains (Fig. 7f), and was indistinguishable from wild-type Mug105 in the RLRGG-AMC cleavage assay (Fig. 7d). As shown in Supplementary Fig. 6d, e, the Mug105[ZUP1-NT] construct was able to interact with both K63-linked di-ubiquitin and longer ubiquitin chains, suggesting that the grafted ubiquitin-binding domains are positioned differently from ZUP1 and TcZUP and therefore did not support K63 cleavage.

## Discussion

With biochemical data available for four ZUFSP family members from three different kingdoms, and structural data available for

three of them, a number of common features defining the family become apparent. A hallmark of ZUFSP proteins is a papain-fold catalytic core domain with the catalytic Cys–His–Asp triad and two conserved acidic residues recognizing the two C-terminal arginine residues of ubiquitin within the RLRGG motif. This core domain is most similar to the UFSP and ATG4 proteases, which target ubiquitin-like modifiers of the UFM1 and Atg8 families, despite the fact that the latter enzymes use a variant Cys-Asp-His active site[13,14]. Except for the minimalistic Mug105 from *S. pombe*, all ZUFSP family members contain a number of N-terminal ubiquitin-binding domains, which are dispensable for catalysis itself, but are required for chain cleavage.

There are a number of properties that set the ZUFSP family apart from other deubiquitinase classes. While many DUBs contain additional ubiquitin-binding domains, the outgoing S1 ubiquitin is typically recognized by the catalytic domain, using an extensive interaction surface covering 20–40% of the bound ubiquitin molecule[3,15]. Mutation of the catalytic cysteine to alanine is able to convert different types of DUB catalytic domains into high-affinity ubiquitin-binding domains[16]. By contrast, the K63-specific ZUFSP family members ZUP1 and TcZUP use the non-catalytic zUBD domain to recognize the Ile-44 patch of the S1 ubiquitin. This UBD, together with the residues recognizing the R-x-R motif, is crucial for chain cleavage. The importance of the UBDs for ubiquitin binding is also demonstrated by pulldown experiments, showing that—unlike full-length ZUP1—the catalytic domain alone is unable to bind ubiquitin[8].

Another hallmark of typical cysteine-type DUBs is the presence of an aromatic "gatekeeper" residue directly following the catalytic histidine. This residue is important for catalysis and helps to exclude larger side chains from the penultimate position of the substrate[12], thereby contributing to the specificity of C-terminal recognition. The aromatic gatekeeper residue is also found in proteases targeting ubiquitin-like modifiers with a C-terminal GG

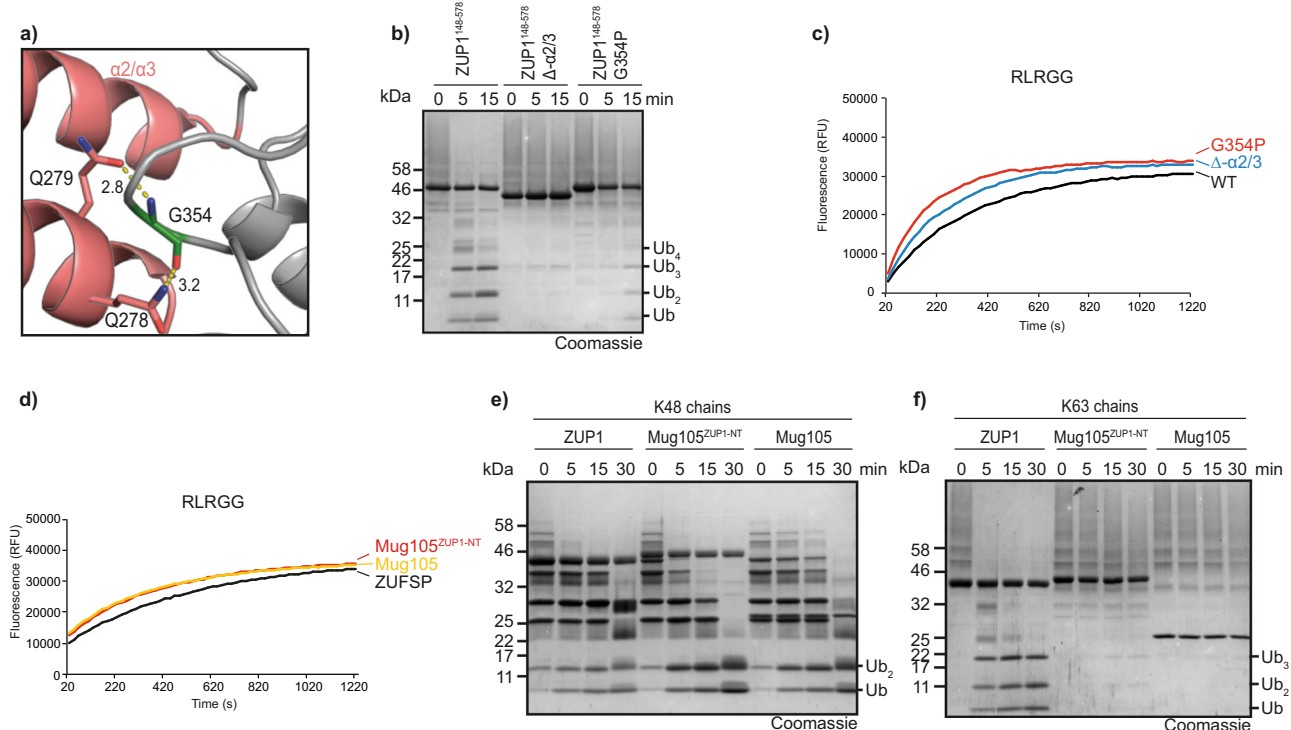

**Fig. 7 Role of the α-2/3 region in chain cleavage. a** The α-2/3 helices are rigidly connected with the catalytic core of ZUP1 (6EI1). The α-2/3 region is shown in cartoon representation and colored red. Gln-278 and Gln-279 contact Gly-354. These key residues are shown as sticks and hydrogen bonds are indicated by yellow dotted lines. **b, c** Activity assays of ZUP1[148–578], the internal truncation ZUP1[148-578Δα-273] and G354P-mutated ZUP1[148–578] against K63-linked Ub[6+] chains (**b**) or RLRGG-AMC (**c**). **d–f** Activity assays of chimeric Mug105[ZUP1-NT] against RLRGG-AMC (**d**), K48-linked ubiquitin chains (**e**), or K63-linked ubiquitin chains (**f**). The chimeric constructs consist of FL Mug105 to which the region 187–319 (ZnF4-MIU-zUBD-α2/3) of human ZUP1 has been N-terminally fused. Source data are provided as a Source Data file.

motif such as SUMO and NEDD8, and has proven useful for the discrimination of GG-directed proteases from other cysteine protease classes[12]. The ZUFSP family is the only eukaryotic DUB class lacking this motif, possibly due to its shared evolutionary ancestry with ATG4 and UFSP proteases, which cleave atypical ubiquitin-like modifiers not ending on GG[14,17]. All ZUFSP family members share a highly conserved tryptophan residue (Trp-421 in ZUP1), which contacts the ubiquitin C terminus and is essential for catalysis (Fig. 5b–g). In this respect, this tryptophan residue resembles the gatekeeper motif of other DUB classes, but its position relative to the active site is different and there is no exclusion of alanine or valine residues from the penultimate position of ubiquitin (Fig. 5h).

A third unusual feature shared between all ZUFSP-type deubiquitinases is the high activity against the peptide substrate RLRGG-AMC, corresponding to the last five residues of ubiquitin. Other DUB classes require a fully-folded S1 ubiquitin for activity, indicating that the recognition of the ubiquitin C terminus is not sufficient. Few DUBs have been shown to also react with RLRGG-AMC, among them UCHL3 and coronaviral PLPro, but their reactivity remains several orders of magnitude below that of ubiquitin-AMC[18,19]. The presence of a complete and properly folded S1 ubiquitin as a prerequisite for DUB cleavage forms the basis of the split-ubiquitin system, a technique for the identification of protein interactions[20]. ZUFSP family members, however, cleave RLRGG-AMC much faster than ubiquitin-AMC[5] (Fig. 2b, c). In accordance with this observation, ZUP1 has been reported as a frequent false-positive in split-ubiquitin experiments[21], suggesting that besides RLRGG-AMC, other N-terminally truncated ubiquitin species are also recognized as substrates. The molecular basis for this preference is not entirely

clear. Our structural and biochemical data show that the last five residues of ubiquitin fit well into the catalytic pocket and are held in place by two salt bridges between the R-*x*-R motif of ubiquitin and conserved acidic residues of the DUB. However, the same is true for many other DUBs that do not act on RLRGG-AMC. It is likely that for conventional DUB:Ub interactions, the R-*x*-R recognition alone is not sufficient and other ubiquitin surfaces have to be recognized as well. In vertebrate species, this complex recognition mode has the additional advantage of limiting cross-reactivity towards ISG15, a ubiquitin modifier that shares the C-terminal RLRGG sequence with ubiquitin[22]. Unlike other DUBs, ZUFSP family members appear to bind the RLRGG peptide sufficiently well for cleavage; the activity increase over Ub-AMC might be due to a faster product release.

The most puzzling observation of this study is the switch of linkage specificity observed between the metazoan K63-specific DUBs ZUP1 and TcZUP on the one hand, and the K48-prefering DUBs Mug105 and AtZUP on the other. Since different ZUFSP family members contain different N-terminal ubiquitin-binding domains, a causal connection to the observed specificities appeared likely and is—at least partially—supported by our data. For human ZUP1, which prefers long K63 chains and cleaves at an internal position (endo mode), the division of labor between the UBDs becomes apparent from the available structures: The zUBD domain recognizes the Ile-44 patch of the S1 ubiquitin, the MIU domain is poised to bind Ile-44 of the S2 ubiquitin, while the UBZ and the more N-terminal zinc fingers are expected to bind to the S3 ubiquitin and possibly additional units of the distal chain[5,7]. The rigid coupling of zUBD and MIU on a single uninterrupted helix supports K63-selectivity within the distal chain, but does not impact the linkage preference at the cleavage

site. The insect protease TcZUP works similarly, but contains a UBZ domain rather than an MIU. Since the UBZ is not rigidly connected to the zUBD (Fig. 3c), a K63-selection within the distal chain is unlikely to happen, leaving only the K63 specificity at the cleavage site. How exactly the recognition of K63-linked S1′ ubiquitin works is less clear. Both ZUP1 and TcZUP contain a structurally conserved helix-turn-helix region (α2/3) positioned at the S1′ binding site. Despite their structural equivalence (Supplementary Fig. 6a), the sequences of the α2/3 regions of ZUP1 and TcZUP are completely dissimilar. A number of point mutations within the ZUP1 α2/3 region did not alter chain cleavage properties, even a quintuple mutant of all plausible interacting residues resulted in an enzyme with wild-type properties (Supplementary Fig. 6c). However, deletion of the α2/3 region or its displacement by the G354P mutation abrogated chain cleavage (Fig. 7b). These results suggest that the α2/3 region acts mainly as a steric block, preventing non-K63-linked ubiquitin chains from fitting into the active site—and possibly also by binding K63-linked chains through main chain contacts. The more distal ubiquitin-binding domains in the N-terminal region of metazoan family members have no measurable influence on the DUB activity against di-ubiquitin or short chains, but appear to cause the long-chain preference seen in ZUP1 and TcZUP, but not in Mug105 and AtZUP (Fig. 3f).

The simplest K48-selective ZUFSP family member is Mug105 from *S. pombe*, which lacks all UBDs, but is nevertheless able to cleave ubiquitin chains. It is likely that this minimalistic enzyme evolved from a UBD-containing ancestor, since enzymes of this type are only observed in narrow fungal lineage that includes *Schizosaccharomyces*. The catalytic core domains of ZUP1 and TcZUP closely resemble the Mug105 structure, but are not able to cleave chains of any linkage type. Thus, Mug105 must have acquired a linkage-selective ubiquitin-binding surface after the secondary loss of its UBDs. Most likely, Mug105 binds and positions the S1 ubiquitin mainly via the C-terminal R-*x*-R motif, since a model of the predicted Mug105:Ub interface shows no other meaningful contacts (Supplementary Fig. 1b). The recognition of the proximal K48-linked S1′ ubiquitin is less clear. Since all attempts of crystallizing Mug105 with di-ubiquitin failed, it can only be speculated that the S1′ ubiquitin is bound by a surface of the core domain, which in other ZUFSP family members is absent or obstructed by UBDs. In the plant protein AtZUP, a UBZ-type zinc finger takes the place of the α2/3 region and also assumes its role in S1′-binding: An AtZUP construct lacking this zinc finger is fully active against the RLRGG-AMC model substrate (Fig. 6c) and reacts with the Ub-PA activity-based probe (Fig. 6d), but does not cleave ubiquitin chains (Fig. 6a, b)—the expected consequence of a defect in S1′ ubiquitin recognition. The use of non-catalytic UBDs for recognition of the S1′ ubiquitin is also observed in other deubiquitinases such as USP5 (IsoT)[23].

Several pieces of evidence suggest that ZUFSP family deubiquitinases are working at sub-optimal speed and are thus unlikely to be involved in bulk deubiquitination within the cell. The activity of human ZUP1 against a model substrate can be increased considerably by a single G514S point mutation (Fig. 4b, e) without strongly affecting chain cleavage, while all activities of plant AtZUP can be increased dramatically by the deletion of a putative auto-inhibitory loop (Fig. 6e). In particular, the latter example makes it likely that AtZUP activity is regulated, either by cleavage within the loop, or by binding of an activator causing a conformational change of the loop region and allowing better access to the active site.

The biological role of the specificity dichotomy is also enigmatic, considering that all ZUFSP family members studied here are singletons in their organisms' proteome and thus are assumed

to fulfill analogous functions. Previous work on human ZUP1 had suggested roles in ssDNA binding, DNA repair and genome stability[5,7,8,24], pathways that are relevant for all species containing a member of the ZUFSP family—but also for species lacking such an enzyme. Moreover, there are no reports of pertinent processes that would require K63 chains in animals, but K48 chains in plants and fungi. Many of the high-confidence interactors of human ZUP1 reported previously have no homologs in insects, fungi or plants and are thus unlikely to mediate activities common to the ZUFSP family[5,7,8]. The unique ability of ZUFSP enzyme to cleave after partial ubiquitin units might indicate a role in cleaving misfolded or fragmented ubiquitin chains. However, the linkage specificity and preference for long chains argue against such a role. Based on currently available data, the most plausible scenario is a neo-functionalization of metazoan ZUFSP proteins within the DNA-damage pathway, which is accompanied by the acquisition of K63 specificity mediated by the zUBD and α-2/3 modules, and of long-chain preference mediated by additional upstream UBDs.

## Methods

**Cloning and mutagenesis.** The ZUP genes were amplified by PCR from *A. thaliana* and *T. castaneum* cDNA (kind gifts from M. Hülskamp and S. Roth, University of Cologne) using the Phusion High Fidelity Kit (New England Biolabs). The PCR fragments were cloned in the pOPIN-S or pOPIN-K vectors[25] using the In-Fusion HD Cloning Kit (Takara Clontech). Point mutations were generated using the QuikChange Lightning kit (Agilent Technologies).

Constructs for ubiquitin-PA purification (pTXB1-ubiquitin1–75) were a kind gift of D. Komander (WEHI, Melbourne). Ub$^{G75A}$ and Ub$^{G75V}$ were amplified from pTXB1-ubiquitin1–75 using primers harboring the respective mutations and cloned in the pTXB1 vector (New England Biolabs) by restriction cloning according to the manufacturers' protocol. Sequences of all oligonucleotides are provided in Supplementary Table 1.

**Protein expression and purification.** All ZUFSP family DUBs including all truncations and mutants were expressed from pOPIN-S vector with an N-terminal 6His-Smt3-tag. Full-length AtZUP was expressed from pOPIN-K vector with an N-terminal 6His-GST-tag. *Escherichia coli* (Strain: Rosetta (DE3) pLysS) were transformed with constructs expressing DUBs and 2–6 l cultures were grown in LB media at 37 °C until the OD600 of 0.8 was reached. The cultures were cooled down to 18 °C and protein expression was induced by the addition of 0.2 mM isopropyl β-D-1-thiogalactopyranoside (IPTG). After 16 h, the cultures were harvested by centrifugation at 5000×*g* for 15 min. After freeze-thaw, the pellets were resuspended in binding buffer (300 mM NaCl, 20 mM TRIS pH 7.5 20 mM imidazole, 2 mM β-mercaptoethanol) containing DNase and Lysozyme, and lysed by sonication using 10 s pulses with 50 W for a total time of 10 min. Lysates were clarified by centrifugation at 50,000×*g* for 1 h at 4 °C and the supernatant was used for affinity purification on HisTrap FF columns (GE Healthcare) according to the manufacturer's instructions. For all constructs, the 6His-Smt3 tag was removed by incubation with Senp1$^{415–644}$ and concurrent dialysis in binding buffer. The liberated affinity-tag and the His-tagged Senp1 protease were removed by a second round of affinity purification with HisTrap FF columns (GE Healthcare). All proteins were purified with a final size exclusion chromatography (HiLoad 16/600 Superdex 75 or 200 pg) in 20 mM TRIS pH 7.5, 150 mM NaCl, 2 mM dithiothreitol (DTT), concentrated using VIVASPIN 20 Columns (Sartorius), flash-frozen in liquid nitrogen, and stored at −80 °C. Protein concentrations were determined using the absorption at 280 nm ($A_{280}$) using the proteins' extinction coefficients derived from their sequences.

**Synthesis of activity-based probes.** All ubiquitin activity-based probes used in this study were expressed as C-terminal intein fusion proteins as described previously[26]. The fusion proteins were affinity purified in buffer A (20 mM HEPES, 50 mM sodium acetate pH 6.5, 75 mM NaCl) from clarified lysates using Chitin Resin (New England Biolabs) following the manufacturer's protocol. On-bead cleavage was performed by incubation with cleavage buffer (buffer A containing 100 mM MesNa (sodium 2-mercaptoethanesulfonate)) for 24 h at room temperature (RT). The resin was washed extensively with buffer A and the pooled fractions were concentrated and subjected to size exclusion chromatography (HiLoad 16/600 Superdex 75 pg) with buffer A. To synthesize Ub-PA, 300 µM Ub-MesNa were reacted with 600 µM propargylamine hydrochloride (Sigma Aldrich) in buffer A containing 150 mM NaOH for 3 h at RT. Unreacted propargylamine was removed by size exclusion chromatography and Ub-PA was concentrated using VIVASPIN 20 Columns (3 kDa cutoff, Sartorius), flash-frozen, and stored at −80 °C. The conversion of Ub-MesNa to Ub-PA was confirmed by intact mass analysis.

**Chain generation**. Met1-linked di-ubiquitin was expressed as a linear fusion protein and purified by ion-exchange chromatography and size exclusion chromatography. K11-, K48-, and K63-linked ubiquitin chains were enzymatically assembled using UBE2S$\Delta$C (K11), CDC34 (K48), and Ubc13/UBE2V1 (K63) as previously described[27,28]. In brief, ubiquitin chains were generated by incubation of 1 µM E1, 25 µM of the respective E2 and 2 mM ubiquitin in reaction buffer (10 mM ATP, 40 mM TRIS (pH 7.5), 10 mM MgCl$_2$, 1 mM DTT) for 18 h at RT. The reaction was stopped by a 20-fold dilution in 50 mM sodium acetate (pH 4.5) and chains of different lengths were separated by cation exchange using a Resource S column (GE Healthcare). Elution of different chain lengths was achieved with a gradient from 0 to 600 mM NaCl.

**Crystallization**. 100 µM TcZUP$^{218-592}$ were incubated with 200 µM ubiquitin-PA for 18 h at 4 °C. Unreacted TcZUP and Ub-PA were removed by size exclusion chromatography. The covalent TcZUP$^{218-592}$/Ub-PA complex and Mug105 (10 mg/ml) were crystallized using the vapor diffusion sitting drop method. Crystallization trials were set up with drop ratios of 1:2, 1:1, 2:1 protein solution to precipitant solution with a total volume of 300 nl.

Initial Mug105 crystals appeared in the conditions SaltRX A1 (1.8 M sodium acetate pH 7.0; 0.1 M BIS-TRIS propane pH 7.0) and JCSG + E4 (0.2 M lithium sulfate; 0.1 M TRIS pH 8.5; 1.26 M ammonium sulfate) after 2 days at 20 °C. Optimization was carried out with 3 µl drops (protein/precipitant ratios: 2:1, 1:1, and 1:2) and precipitant solutions varying in pH or sodium acetate concentration respectively. Optimized crystals were harvested and cryoprotected with a reservoir containing a final concentration of 2.6 M sodium acetate.

Initial TcZUP$^{218-592}$/Ub-PA complex crystals appeared in a series of conditions. The most promising crystals appeared in Index H2 (0.2 M potassium sodium tartrate; 20% (w/v) PEG3350) and Peg/Ion F4 (8% (v/v) Tacsimate pH 7.0; 20% (w/v) PEG3350); hence these conditions were selected for further optimization. The best data set was obtained from a crystal harvested from the optimization of Peg/Ion F4 (8% (v/v) Tacsimate pH 7; 22% PEG3350) and cryoprotected with reservoir solution supplemented with 18% ethylene glycol.

**Data collection, phasing, model building, and refinement**. Diffraction data for the TcZUP$^{218-592}$/Ub-PA complex were collected at beamline X06DA/PXIII at Swiss Light Source (SLS) (Paul Scherrer Institut, Villigen, Switzerland) and processed using XDS[29]. The structure was solved by molecular replacement using a truncated model of the ZUP1 in complex with ubiquitin (residues 254–578 of ZUFSP and all residues of ubiquitin; PDB:6EI1) as search model and phenix.phaser as molecular replacement program[30]. Afterward, the model was built using the phenix.autobuild routine followed by iterative cycles of manual model building using coot and refinements using phenix.refine[31,32]. Restrains of the propargyl moiety, ethylene glycol, and citrate were calculated using phenix.elbow[33].

Diffraction data for Mug105 were collected at beamline ID30B, ESRF, Grenoble, France and processed using XDS[29]. The crystal structure was solved with by molecular replacement using a truncated polyalanine model of the catalytic domain of ZUP1 (residues 321–578) as searching model and phenix.phaser as molecular replacement program[30]. Afterward, the model was built using the phenix.autobuild routine followed by iterative cycles of the manual model building using coot and refinements using phenix.refine[31,32].

**AMC assays**. Activity assays of DUBs against AMC-labeled substrates were performed using reaction buffer (150 mM NaCl, 20 mM TRIS pH 7.5, 10 mM DTT), 1 µM DUBs, 100 µM zRLRGG-AMC (BACHEM AG, Switzerland) or 5 µM Ub-AMC (UbiQ-Bio, The Netherlands). The reaction was performed in black 96-well plates (Corning) at 30 °C and fluorescence was measured using the Infinite F200 Pro plate reader (Tecan) equipped for an excitation wavelength of 360 nm and an emission wavelength of 465 nm and the data was collected using Magellan 7.1 software (Tecan). The presented results are means of three independent cleavage assays.

**Kinetics**. Steady-state kinetics of ZUP1 or TcZUP against RLRGG-AMC were measured in reactions containing either 100 nM ZUP1 or 20 nM TcZUP and the indicated concentrations of RLRGG-AMC in reaction buffer (150 mM NaCl, 20 mM TRIS pH 7.5, 10 mM DTT). Measurements were performed at 30 °C in triplicate. Initial velocities were plotted against the RLRGG-AMC concentrations and fitted to the Michaelis–Menten equation using Prism 6 (GraphPad) software.

**Activity-based probe assays**. DUBs were prediluted to 2× concentration (10 µM) in reaction buffer (20 mM TRIS pH 7.5, 150 mM NaCl and 10 mM DTT) and 1:1 combined with 100 µM Ub-, Ub$^{G75A}$-, and Ub$^{G75V}$-PA. After 3 h incubation at RT, the reaction was stopped by the addition of 2× Laemmli buffer, resolved by SDS-PAGE, and Coomassie-stained. At least two technical replicates were performed and one representative gel is shown.

**Ubiquitin chain cleavage**. DUBs were preincubated in 150 mM NaCl, 20 mM TRIS pH 7.5 and 10 mM DTT for 10 min. The cleavage was performed for the indicated time points with 5 µM DUBs and either 25 µM di-ubiquitin (K11, K48,

K63, and M1 synthesized as described above, others from Boston Biochem) or 5 µM tetra-ubiquitin (Boston Biochem) at RT. After the indicated time points, the reaction was stopped with 2× Laemmli buffer, resolved by SDS-PAGE, and Coomassie-stained. At least two technical replicates were performed and one representative gel is shown.

**Ubiquitin-binding assay**. A total of 10 µl glutathione particles (MagneGST™ Glutathione Particles, Promega) were saturated with 6His-GST tagged Mug105 or Mug105$^{ZUP1-NT}$ in 200 µl binding buffer (20 mM TRIS pH 7.5, 150 mM NaCl, 20 mM imidazole, and 0.1% NP-40) and incubated for 1 h at 4 °C. Both proteases were inactivated by a C42A mutation. The particles were washed three times with binding buffer and afterward incubated with the twofold molar excess of K63-linked ubiquitin chains for 2 h at 4 °C. The washing steps were repeated and the protein was eluted from the beads by addition of 30 µl Laemmli buffer. The proteins were separated via SDS-PAGE and visualized by coomassie staining or immunostaining with α-ubiquitin P4D1 antibody (1:3000; 05-944; EMD Millipore Corp).

**Intact mass analysis**. Purified AtZUP was diluted to 1 µg/µl in 0.1% formic acid and analyzed on a Q Exactive Plus Orbitrap (Thermo Scientific) mass spectrometer that was coupled to an EASY nLC (Thermo Scientific). Proteins were diluted in solvent A (0.1% formic acid in water) and loaded onto an in-house packed pulled tip column (18 cm length, 75 µm I.D., filled with MABPac™ RP polymeric resin, 4 µm, 1.500 Å, Thermo Scientific). The column was operated at 50 °C and proteins were separated at a constant flow rate of 600 nL/min using the following gradient: 3-15% solvent B (0.1% formic acid in 80 % acetonitrile) within 1.0 min, 15-60% solvent B within 15.0 min, 60–95% solvent B within 1.0 min, followed by washing and column equilibration. Full MS1 scans were acquired from 600 to 2000 m/z combining 2 microscans at a resolution of 140,000. The AGC target was set to 3E6 and the maximum injection time to 200 ms. Combined exact mass spectra of selected scans were exported from Excalibur 3.1 (Thermo Scientific). Annotation of monoisotopic masses and subsequent deconvolution of charge clusters was done in mMass 5.5[34].

**Statistics and reproducibility**. All activity-based probes, chain cleavage, ubiquitin-binding, and AMC assays were performed two independent times with similar results. Each AMC assay was additionally performed in triplicates for noise reduction. The intact mass analysis was performed once.

**Reporting summary**. Further information on research design is available in the Nature Research Reporting Summary linked to this article.

## Data availability

The X-ray structures of Mug105 and TcZUP::Ub-PA generated in this study have been deposited at the PDB database under the accession numbers 7OIY and 7OJE, respectively. The X-ray structure of ZUP1 is publicly available at the PDB database under the accession number 6EI1. The data underlying the findings of this study are available in this article and its Supplementary Information or are available from the corresponding author upon reasonable request. Source data are provided with this paper.

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

## Acknowledgements

We thank Christiane Horst for expert technical assistance, Siegfried Roth for a kind gift of *T. castaneum* RNA, Martin Hülskamp for a kind gift of *A. thaliana* RNA and Monique Mulder (Leiden University) for a sample of Nedd8-PA. We thank the staff of the CECAD proteomics facility for their support. We thank the staff of beamline X06DA at the Swiss Light Source, Paul Scherrer Institute, Villigen (Switzerland), and beamline ID30B at the European Synchrotron Radiation Facility (ESRF), Grenoble (France) for their support during data collection. Deubiquitinase research in the lab of KH is supported by DFG grant HO 3783/3-1. Crystals were grown using equipment of the Cologne Crystallization facility (C$_2$f), which is supported by DFG grant INST 216/949-1 FUGG.

## Author contributions

T.H. performed most experiments, C.P. solved the X-ray structures, U.B. supervised the crystallography, K.H. initiated and supervised the study, and contributed bioinformatical analyses. All authors contributed to data analysis and the writing of the manuscript.

## Funding

## Competing interests

The authors declare no competing interests.
