## [Peer Review File · Nature Communications]

REVIEWER COMMENTS

Reviewer #1 (Remarks to the Author):

The authors have described their attempts at understanding structural basis for the diverse linkage-specific polyubiquitin recognition in a select group of enzymes belonging to the recently described ZUFSP family of DUBs, the 7th and the last DUB family to be added to the list of eukaryotic deubiquitinases. The ZUFSP family, with a single mammalian member (human ZUP1, for example), is structurally distinct from the other six cysteine protease DUB families, appears to be more closely related in evolution to enzymes that deconjugate ubiquitin-like modifies, such as UFM1 and ATG8, than ubiquitin-specific hydrolases. Yet, homologs of ZUP1 have acquired DUB activity at the expense of ubiquitin-like deconjugase activity. In light of this evolutionary background, polyubiquitin linkage specificity/preference of ZUFSP members is an interesting subject for investigation. Insights gathered from this line of investigation will likely expand our understanding of ubiquitin biology in general.

In a 2018 paper introducing the ZUFSP family, the same group described their discovery of the ZUFSP family of DUBs, demonstrating the biochemical properties of Zup1 and solving the crystal structure of a construct of Zup1 bound to the ubiquitin-based covalent inhibitor Ub-PA. Among the main biochemical and structural results to emerge from that report was the description of ZUP1 as a modular enzyme with a catalytic core appended with a number of Ub-interacting domains some of which conferred K63-linkage specific recognition (zUBD and α 2- α 3 domains) and others contributed length-dependent activity by recognizing additional Ub monomers distal to the cleavage site. They also introduced *S. pombe* homolog of Zup1, Mug105, which represents a minimalist version of ZUFSP in that it contains just the catalytic core lacking other ubiquitin recognizing modules.

In this manuscript the authors have embarked on an interesting analysis by selecting Zup1 homologs from an insect, plant, fungi species and comparing their properties to human Zup1. As a premise of this comparison, the authors argue that the most reduced version of the enzyme (Mug105) possesses K48-linked polyubiquitin preference as a 'default' activity, which was altered to K63-linkage specificity with acquisition of some specific combination of ubiquitin-binding domains (UBDs). Indeed, the group of enzymes they chose to study do exhibit interesting differences in the composition of domains relative to the catalytic core. The authors have recognized these UBDs using bioinformatics and shown their arrangement in the primary structures of these enzymes. Structure of full-length Mug105 was solved to 2 Å, which confirmed that it is a true homolog of human Zup1. Also solved was the Ub-PA bound structure of an insect Zup1 homolog, AcZup. The AcZup construct revealed a similar arrangement of UBDs, including binding of the S1-Ub, as the human enzyme, leading the authors to conclude that the presence of a particular combination of two UBDs confers Lys63-linkage preference to metazoan ZUFSPs. The structural studies were performed with expected rigor and the overall quality of the biochemical data are also good.

I would urge the authors to consider the following points to improve the manuscript.

1) At one point the authors claim that S1-binding site of Mug105 does not depend on Ile-44 recognition by the zUBD domain, but otherwise similar to that of human Zup1. This statement implies that the Ile44 patch of S1-Ub is not involved in Mug105 substrate recognition. This needs to be examined further. One can test this using the I44A mutant of Ub-PA. It might help to generate a model of Ub or K48-diUb bound to Mug105. This sort of results will complement the Mug105 structural data by providing relevant information on Ub binding, which is sorely missing in the paper as the authors have not managed to crystallize Mug105 with Ub. From their description it appears that Mug105 recognizes Ub primarily by engaging the C-terminal tail while the rest of S1-Ub, especially the crucial I44 patch, remain solvent exposed.

2) Oxyanion hole: This section is a bit confusing, does not seem to fit with the rest of the manuscript and it needs further clarification. While it is somewhat interesting that placement of a Ser (in place of Gly) in the active-site pocket of Zup1 results in higher activity in the peptide hydrolysis assay, the effect in polyubiquitin chain is rather marginal. The gain in activity toward the peptide substrate could be a substrate binding effect (K_m effect) rather than a k_{cat} effect, which is what one would expect of an oxyanion stabilizer. The authors should provide kinetic parameters of the serine mutant (and the WT enzyme) using the peptide substrate to substantiate their claim of oxyanion stabilization in the mutant.

3) Since the authors propose Zup1 as a modular enzyme, it would be interesting to see if the zUBD-~~22-23~~ construct alone (outside the context of the catalytic domain) retains K63-linkage preference (binding to diubiquitin chains). I think this will strengthen the argument that these domains together serve as a determinant of K63-linkage preference. The authors have attempted to graft these domains into Mug105 in the hopes of producing K63 preference in that chimeric construct. Does that construct bind to K63 diub, even though it failed to hydrolyze it?

4) It seems when the C-terminal Ub peptide is presented as a part of Ub, instead of being the peptide alone (LRLRGG-AMC substrate), the hydrolytic activity is much reduced. Does that mean there are inhibitory elements in ZUFSP enzymes that occlude Ub binding? Based on their structural results can the authors get any answer to this riddle?

5) Please include NEDD8-PA, SUMO-PA, LC3-PA modification data for all enzymes studied here (like Fig S1B).

Minor Points:

1. In Figure 2A, please indicate the name of the Zup1 homolog next to organism name.
2. Figure 2B, there seems to be some mislabeling in the figures (ArathZ and TriboZ). Please use AtZup and TcZup as described in the legend and in the main text.
3. Why is K27 poly-Ub missing in the assays?
4. Explain the red and green arrows used in Figure 2 in the associated legend.
5. Why was GST-AtZUP1 used in the polyubiquitin chain cleavage assay?
6. For the TcZUP286-592 construct, please check activity using the Ub-derived peptide substrate to rule out folding defects of the deletion mutant.

Reviewer #2 (Remarks to the Author):

In this manuscript, Hermanns et al analyse the roles of non-catalytic domains of ZUFSP in polyubiquitin cleavage by studying five different ZUFSP family members from human, insects, fungi and plants. Their analyses reveal that domains at the N-terminus of the protein is important for activity and linkage preference. Interestingly, they find that ZUFSP intrinsically has K48 linkage preference and the additional ubiquitin binding domains (UBDs) change this enabling the enzyme to cleave K63 chains. By determining structures of insect ZUFSP, they identify that human ZUP1 has a sub-optimal oxyanion hole that likely explains its low activity. Overall, this is a systematic study that provides detailed insights into how polyUb chains are processed by ZUFSP in different species. All experiments have been performed with adequate controls. I only have a few minor concerns:

Minor points:

1. We showed in our study (Kwasna et al 2018) that the 4th ZnF is a UBZ domain that binds polyUb and is essential for ZUP1 to cleave K63 chains. It would be fair to mention this in lines 59-60
2. In figure 1, ubiquitin from the ZUP1 structures could be superposed to show the reader where Ub would be positioned in the Mug105 structure

3. Figure 2 D and F: could the authors please state in figure legends why the gels have been spliced. Ideally, these should be on the same gel
4. Fig 2 also has 'bioinformatics?' at the top of the page
5. Figure 3A could be labelled to show which one is Ub, UBZ, etc
6. It would be helpful to see some quantification for assays like that in Figure 6E, 6G, 7E where activity of different mutants is compared. This would reveal if the differences are significant.
7. Please specify on the figures, which chain linkage is being assayed. This would make it easier.

Yogesh Kulathu

Reviewer #3 (Remarks to the Author):

In this manuscript Hermanns et al. describe the diverse linkage specificities within the novel ZUFSP deubiquitinase family. In particular they characterize the Ub chain preference of ZUFSP members from distant organisms, and they ascribed this preference to the presence of different N-terminal domains. They also solve the crystal structure of two members of the ZUFSP family, from yeast and from insect, the latter in complex with ubiquitin-PA, and compare the structures to the first reported hZUP1. The structures and functional characterization of ZUFSP family members presented in this manuscript are interesting and add more information to the knowledge of this novel type of DUB. However, the conclusions after the structural and functional analysis are unsatisfactory for quality and scope of the journal, since they do not conclude to explain the Ub chain preference by the N-terminal UBD domains. The authors claim the relevance of the α 2-3 domain for the K63 preference, but this should be better confirmed with more evidence.

Please find below some suggestions to improve the quality of the work:

Major Points:

The authors first show the crystal structure of Mug105 from *S.pombe*, which is a minimal version of ZUP1 containing only the catalytic papain-like domain that cleaves Ub K63-linkages. The structure is similar to hZUP1. I expected a better explanation for the Ub K48 preference in the absence of any N-terminal UBD domain and having only with the RxR motif. For example, an analysis of Ub linkage preferences for Mug105 would be interesting (as done later for insect and plant).

Minor points: In Figure 1B, are the authors sure of the side chain conformer of His165 ???. Shouldn't the polar nitrogen be oriented towards the cysteine ?. Fig 1F should be better labeled, what are all the bands ? K48 chains ?. Also in fig1K. Better show in Fig1i that Ub chains are K63, otherwise it is confusing. In line 61 of the main text, *Saccharomyces pombe* does not exist (please correct).

The authors next analyze the Ub chain preference of the insect and plant ZUFSP family members, that contain different N-terminal modules. Insect prefers K63, while plant prefers K48 chains. The authors solved the structure of insect tcZUP with Ub-PA. The complex structure resembles human ZUP1, with a similar Ub interface to zUBD, but contains a unique UBZ-like Zn-finger domain. The authors show the role of the N-terminal Zn-domains in the activity of K63 UB chains. They show a loss of activity, but nothing about a possible change in the Ub chain preference ?. This should be a major point in the study, where does the preference for K63 comes from ?. Direct binding of K63 chains with Zn finger domains??. Last gel in Fig 3F is confusing (where is Ub4 b and if it is not cleaved). Also, it would be interesting to check the activities of the truncated mutants against diUB-K63 substrate, I would expect cleavage only with the presence of a2-3 and ZUBD (each domain for the proximal and distal ubiquitin). This would confirm the role of a2-3 in the K63 preference, and is a major point in the manuscript.

Minor points: Are the proteins well purified ? There are many extra bands in the SDS-PAGE of figure 2. The authors should consider a better labeling of the gels in Fig 2. The color of protein domains in the structure cartoons in Fig 3 should match the domain scheme.

The authors checked next the composition of the oxyanion hole, and say that a serine in that position enhances the catalytic activity of hZUP1 over glycine; and they also check the role of a tryptophan as a gatekeeper in the ubiquitin tail recognition. Although informative, it does not add anything to the substrate preference analysis.

Minor points: Again SDS-PAGE gels should be better labeled. The reaction of ZUP1 with Ub-PA is quite poor in figure 5h. I would expect a better yield.

The N-terminal Zn finger in plants is essential for K48 cleavage, in contrast to Mug105 which has K48 preference without any extra domain (the gel in Fig6a is ugly and should be redone). Finally, the role of the a2/3 is analyzed in ZUP1, which might be the essential interface for K63 chain preference of the ZUFSP members. However, the results with the quimeric Mug105 with ZUP1 N-terminal domain elements are discouraging, and a change in the specificity is not observed.

All the results confirm the Ub chain substrate preferences for the different ZUFSP members, but do not shed light to the structural mechanisms to reach chain preferences for each particular member. Only the $\alpha 2/3$ domain seems necessary for K63 cleavage. It would be really interesting a complex structure with a diUb-K63 substrate. Also, I would expect a better structural-based explanation for the Ub linkage preference in the plain Mug105 domain. What elements in the deubiquitinase domain are involved in chain preference ?.

Reviewer #1 (Remarks to the Author):

The authors have described their attempts at understanding structural basis for the diverse linkage-specific polyubiquitin recognition in a select group of enzymes belonging to the recently described ZUFSP family of DUBs, the 7th and the last DUB family to be added to the list of eukaryotic deubiquitinases. The ZUFSP family, with a single mammalian member (human ZUP1, for example), is structurally distinct from the other six cysteine protease DUB families, appears to be more closely related in evolution to enzymes that deconjugate ubiquitin-like modifies, such as UFM1 and ATG8, than ubiquitin-specific hydrolases. Yet, homologs of ZUP1 have acquired DUB activity at the expense of ubiquitin-like deconjugase activity. In light of this evolutionary background, polyubiquitin linkage specificity/preference of ZUFSP members is an interesting subject for investigation. Insights gathered from this line of investigation will likely expand our understanding of ubiquitin biology in general.

In a 2018 paper introducing the ZUFSP family, the same group described their discovery of the ZUFSP family of DUBs, demonstrating the biochemical properties of Zup1 and solving the crystal structure of a construct of Zup1 bound to the ubiquitin-based covalent inhibitor Ub-PA. Among the main biochemical and structural results to emerge from that report was the description of ZUP1 as a modular enzyme with a catalytic core appended with a number of Ub-interacting domains some of which conferred K63-linkage specific recognition (zUBD and $\alpha 2$ - $\alpha 3$ domains) and others contributed length-dependent activity by recognizing additional Ub monomers distal to the cleavage site. They also introduced *S. pombe* homolog of Zup1, Mug105, which represents a minimalist version of ZUFSP in that it contains just the catalytic core lacking other ubiquitin recognizing modules.

In this manuscript the authors have embarked on an interesting analysis by selecting Zup1 homologs from an insect, plant, fungi species and comparing their properties to human Zup1. As a premise of this comparison, the authors argue that the most reduced version of the enzyme (Mug105) possesses K48-linked polyubiquitin preference as a 'default' activity, which was altered to K63-linkage specificity with acquisition of some specific combination of ubiquitin-binding domains (UBDs). Indeed, the group of enzymes they chose to study do exhibit interesting differences in the composition of domains relative to the catalytic core. The authors have recognized these UBDs using bioinformatics and shown their arrangement in the primary structures of these enzymes. Structure of full-length Mug105 was solved to 2 Å, which confirmed that it is a true homolog of human Zup1. Also solved was the Ub-PA bound structure of an insect Zup1 homolog, AcZup. The AcZup construct revealed a similar arrangement of UBDs, including binding of the S1-Ub, as the human enzyme, leading the authors to conclude that the presence of a particular combination of two UBDs confers Lys63-linkage preference to metazoan ZUFSPs. The structural studies were performed with expected rigor and the overall quality of the biochemical data are also good.

I would urge the authors to consider the following points to improve the manuscript.

1) At one point the authors claim that S1-binding site of Mug105 does not depend on Ile-44 recognition by the zUBD domain, but otherwise similar to that of human Zup1. This statement implies that the Ile44 patch of S1-Ub is not involved in Mug105 substrate recognition. This needs to be examined further. One can test this using the I44A mutant of Ub-PA. It might help to generate a model of Ub or K48-diUb bound to Mug105. This sort of results will complement the Mug105 structural data by providing relevant information on Ub binding, which is sorely missing in the paper as the authors have not managed to crystallize Mug105 with Ub. From their description it appears that Mug105 recognizes Ub primarily by engaging the C-terminal tail while the rest of S1-Ub, especially the crucial I44 patch, remain solvent exposed.

We addressed this issue by i) generating a model of mono-Ub binding to Mug105, and ii) by performing the requested experiment using the I44A mutant of Ub-PA. As shown in the new

supplementary figure S1b, the ubiquitin was modelled into the Mug105 structure by superposition with the ZUP1:Ub-PA covalent complex. The Ile-44 patch, which in ZUP1 is bound by the zUBD helix (light grey) does not form a contact with Mug105. Supplementary figure S1b shows that the I44A mutant of Ub-PA is not impaired for reaction with Mug105 – it even reacts somewhat better than wildtype Ub-PA. These findings support the idea that Ile-44 of ubiquitin is not involved in Mug105 substrate recognition. A description of these findings was inserted on page 4 of the manuscript.

2) Oxyanion hole: This section is a bit confusing, does not seem to fit with the rest of the manuscript and it needs further clarification. While it is somewhat interesting that placement of a Ser (in place of Gly) in the active-site pocket of Zup1 results in higher activity in the peptide hydrolysis assay, the effect in polyubiquitin chain is rather marginal. The gain in activity toward the peptide substrate could be a substrate binding effect (K_m effect) rather than a k_{cat} effect, which is what one would expect of an oxyanion stabilizer. The authors should provide kinetic parameters of the serine mutant (and the WT enzyme) using the peptide substrate to substantiate their claim of oxyanion stabilization in the mutant.

To find out if the increased activity of the human G514S mutant is due to k_{cat} or K_m , we measured kinetic data for wildtype ZUP1, G514S-mutated ZUP1, and wildtype TcZUP. The results have been added to Suppl. Figure S4. The G514S mutant increases k_{cat} of the human enzyme by a factor of 3.4, while K_m is only reduced by 30%. This result supports the idea that G514S is not crucial for substrate binding and rather has a kinetic role compatible with stabilizing the oxyanion hole. A comparison to the *Tribolium* protein TcZUP shows that the latter enzyme has a 14.5-fold higher k_{cat} than human ZUP1, with similar K_m values. Apparently, TcZUP has additional properties increasing catalysis beyond the serine at position 528. We have added the description of the experiments to the result section 'role of the oxyanion hole'.

3) Since the authors propose Zup1 as a modular enzyme, it would be interesting to see if the zUBD- $\alpha 2$ - $\alpha 3$ construct alone (outside the context of the catalytic domain) retains K63-linkage preference (binding to diubiquitin chains). I think this will strengthen the argument that these domains together serve as a determinant of K63-linkage preference. The authors have attempted to graft these domains into Mug105 in the hopes of producing K63 preference in that chimeric construct. Does that construct bind to K63 diub, even though it failed to hydrolyze it?

We performed the suggested experiment and did a pull-down analysis of K63-lined ubiquitin chains (dimer and mixture of 6+ chains) by wildtype Mug105 and the 'grafted' version Mug105^{ZUP1-NT}. As shown in the newly added Suppl. Figure 6d (Coomassie-stained PAGE) and 6e (anti-ubiquitin Western blot), the putative K63-binding domain of ZUP1 grafted onto Mug105 did confer K63-binding for short and long chains, but was not very effective. We consider it most likely that this region is not a strong K63-binder per se, but contributes to ubiquitin binding by ZUP1 and TcZUP1 in the right context, i.e. when it is positioned correctly to work together with the RxR recognition by the catalytic core. We have added a description of the results on page 10 and also address them in the discussion.

4) It seems when the C-terminal Ub peptide is presented as a part of Ub, instead of being the peptide alone (LRLRGG-AMC substrate), the hydrolytic activity is much reduced. Does that mean there are inhibitory elements in ZUFSP enzymes that occlude Ub binding? Based on their structural results can the authors get any answer to this riddle?

Unfortunately, our structural data do not shed light on this question. Inhibitory elements are unlikely to play a role, since the effect is also observed for Mug105 with its minimalistic catalytic domain. One

possibility would be a slower product dissociation rate for ubiquitin due to additional binding contacts. We have added a short paragraph to the discussion section, where we address this puzzle.

5) Please include NEDD8-PA, SUMO-PA, LC3-PA modification data for all enzymes studied here (like Fig S1B).

We have performed the requested experiments and added the results to Suppl. Figure S2 panels h,i (formerly S1). After long (6h) incubation with the activity-based probes, all studied members of the ZUFSP family react with Ub-PA and to a lesser degree with NEDD8-PA, but not with SUMO-PA or LC3-PA. The description of these results has been added to the result section on page

Minor Points:

1. In Figure 2A, please indicate the name of the Zup1 homolog next to organism name.

Figure 2a has been changed accordingly.

2. Figure 2B, there seems to be some mislabeling in the figures (ArathZ and TriboZ). Please use AtZup and TcZup as described in the legend and in the main text.

In the initial submission, we had inadvertently included a preliminary version of figure 2. We have now replaced it by the correct version, which does not contain these mistakes.

3. Why is K27 poly-Ub missing in the assays?

We did not include K27 chains because we had difficulties obtaining this reagent. We have now found a supplier and have added cleavage experiments with K27 di-ubiquitin to Supplementary figure S2e. As expected, no K27-activity was observed for any of the tested enzymes.

4. Explain the red and green arrows used in Figure 2 in the associated legend.

The new & correct version of figure 2 does not contain these arrows anymore.

5. Why was GST-AtZUP1 used in the polyubiquitin chain cleavage assay?

As described in the manuscript, bacterially expressed AtZUP1 is cleaved to a variable degree by bacterial proteases within the flexible loop, giving rise to inhomogeneous preparations with multiple AtZUP1 bands (documented in figure 6a,d and the text). The GST-AtZUP1 construct was less prone to internal cleavage by bacterial proteases and was therefore used for the linkage-specificity panel in figure 2d,e. Note that the same K48 specificity is also seen for a GST-less construct in figure 6g, demonstrating that activated AtZUP also prefers K48 chains.

6. For the TcZUP286-592 construct, please check activity using the Ub-derived peptide substrate to rule out folding defects of the deletion mutant.

We have performed the experiment and added the result to Suppl. Figure S3d The TcZUP²⁸⁶⁻⁵⁹² construct was fully active against the peptide substrate, ruling out a folding defect. We have also included the shorter construct TcZUP³²⁵⁻⁵⁹², which was also fully active against the peptide.

Reviewer #2 (Remarks to the Author):

In this manuscript, Hermanns et al analyse the roles of non-catalytic domains of ZUFSP in polyubiquitin cleavage by studying five different ZUFSP family members from human, insects, fungi and plants. Their analyses reveal that domains at the N-terminus of the protein is important for activity and linkage preference. Interestingly, they find that ZUFSP intrinsically has K48 linkage

preference and the additional ubiquitin binding domains (UBDs) change this enabling the enzyme to cleave K63 chains. By determining structures of insect ZUFSP, they identify that human ZUP1 has a sub-optimal oxyanion hole that likely explains its low activity. Overall, this is a systematic study that provides detailed insights into how polyUb chains are processed by ZUFSP in different species. All experiments have been performed with adequate controls. I only have a few minor concerns:

Minor points:

1. We showed in our study (Kwasna et al 2018) that the 4th ZnF is a UBZ domain that binds polyUb and is essential for ZUP1 to cleave K63 chains. It would be fair to mention this in lines 59-60

We apologize for not having cited this paper initially; it has been added now.

2. In figure 1, ubiquitin from the ZUP1 structures could be superposed to show the reader where Ub would be positioned in the Mug105 structure

We have added the requested superposition to the new Supplementary figure S1b and added a description to the result section on page 4.

3. Figure 2 D and F: could the authors please state in figure legends why the gels have been spliced. Ideally, these should be on the same gel

Our gel system only supports 15 slots per gel. The experiments for figure 2d and 2f involved 21 samples each and therefore did not fit onto a single gel. Please note that we did not 'splice' any gels (something we would consider problematic) but rather show the two gels belonging to a single experiment with individual borders and a gap in between! We intentionally did not cut away the high-MW region, thus allowing to control for equal protease loading.

4. Fig 2 also has 'bioinformatics?' at the top of the page

In the initial submission, we had inadvertently used a preliminary version of figure 2. We have now replaced it by the correct version, which does not contain this mistake.

5. Figure 3A could be labelled to show which one is Ub, UBZ, etc

We have added the requested labels to figure 3a.

6. It would be helpful to see some quantification for assays like that in Figure 6E, 6G, 7E where activity of different mutants is compared. This would reveal if the differences are significant.

We agree that a quantification of SDS-PAGE band intensities is often helpful and sometimes crucial. However, in the present case we are dealing with heterogeneous chain mixtures (Ub₆₊) as substrates, whose quantification is difficult, if at all possible. With regard to figure 6e, we have changed the text (page 10) and now only compare the wildtype AtZUP with the loop-deleted mutations. This activity difference is very obvious and does not rely on quantification. Figure 6g shows a linkage assay and does not compare any mutants, it is not clear how a quantification would help here. Figure 7e is used to document that the mutated construct is still able to cleave K48 chains. The validity of this claim can easily be verified without any quantitative analysis.

7. Please specify on the figures, which chain linkage is being assayed. This would make it easier.

We have added this information (which was originally only contained in the figure legend) to all figure displays, where it helps to clarify the results.

Yogesh Kulathu

Reviewer #3 (Remarks to the Author):

In this manuscript Hermanns et al. describe the diverse linkage specificities within the novel ZUFSP deubiquitinase family. In particular they characterize the Ub chain preference of ZUFSP members from distant organisms, and they ascribed this preference to the presence of different N-terminal domains. They also solve the crystal structure of two members of the ZUFSP family, from yeast and from insect, the latter in complex with ubiquitin-PA, and compare the structures to the first reported hZUP1. The structures and functional characterization of ZUFSP family members presented in this manuscript are interesting and add more information to the knowledge of this novel type of DUB. However, the conclusions after the structural and functional analysis are unsatisfactory for quality and scope of the journal, since they do not conclude to explain the Ub chain preference by the N-terminal UBD domains. The authors claim the relevance of the α 2-3 domain for the K63 preference, but this should be better confirmed with more evidence.

Please find below some suggestions to improve the quality of the work:

Major Points:

The authors first show the crystal structure of Mug105 from *S.pombe*, which is a minimal version of ZUP1 containing only the catalytic papain-like domain that cleaves Ub K63-linkages. The structure is similar to hZUP1. I expected a better explanation for the Ub K48 preference in the absence of any N-terminal UBD domain and having only with the RxR motif. For example, an analysis of Ub linkage preferences for Mug105 would be interesting (as done later for insect and plant).

Sorry for being not sufficiently clear on this: the analysis of Mug105 catalytic properties has of course been done, but was already included in our 2018 ZUFSP paper (Hermanns et al, Nat. Comms 9:799, cited as reference 5 in this manuscript.) In brief, Mug105 selectively cleaves K48 chains and shows no preference for long chains. We have added a sentence to the result section (page 3) to make the reader aware of this fact.

Minor points: In Figure 1B, are the authors sure of the side chain conformer of His165 ?? Shouldn't the polar nitrogen be oriented towards the cysteine ?.

The reviewer is absolutely correct. We had inadvertently used an image from a preliminary refinement of the structure. We have corrected this mistake, figure 1b now shows the correct conformer.

Fig 1F should be better labeled, what are all the bands ? K48 chains ?. Also in fig1K. Better show in Fig1i that Ub chains are K63, otherwise it is confusing. In line 61 of the main text, *Saccharomyces pombe* does not exist (please correct).

We have corrected all of these mistakes and unclarities. The high-MW bands in Figure 1f and 1k are polymeric K48 chains, we have indicated this in the revised figure. We have also corrected the spelling of *Schizosaccharomyces pombe*.

The authors next analyze the Ub chain preference of the insect and plant ZUFSP family members, that contain different N-terminal modules. Insect prefers K63, while plant prefers K48 chains. The authors solved the structure of insect tcZUP with Ub-PA. The complex structure resembles human ZUP1, with a similar Ub interface to zUBD, but contains a unique UBZ-like Zn-finger domain. The authors show the role of the N-terminal Zn-domains in the activity of K63 Ub chains. They show a loss of activity, but nothing about a possible change in the Ub chain preference ?. This should be a

major point in the study, where does the preference for K63 comes from ? Direct binding of K63 chains with Zn finger domains??.

This is a good point, which we have addressed by a number of experiments. For testing if the linkage specificity of TcZUP is changed by truncating the UBDs, we have tested the truncations TcZUP²⁵⁷⁻⁵⁹² and TcZUP²⁸⁶⁻⁵⁹² for cleavage of differently linked di-ubiquitin species. As shown in Suppl. Figure S3e, the TcZUP²⁵⁷⁻⁵⁹² construct was less active than the full-length TcZUP (figure 3f) but maintained the preference for K63 chains. By contrast, the even shorter construct TcZUP²⁸⁶⁻⁵⁹² was mostly inactive against all chain types, even when prolonging the reaction time to 60 minutes. These experiments are now described on page 7.

The question for the basis of K63 specificity in ZUP1 and TcZUP has a complex answer; we have expanded the section in the discussion addressing this point (page 12ff). In brief, our results support a model in which K63 specificity requires i) a robust binding of the S1-ubiquitin (beyond mere RxR recognition), and ii) an S1'-ubiquitin positioning that is compatible with the alpha-2/3 region without using sequence-specific recognition. The first requirement involves the proximal UBDs and is demonstrated by the inability of UBD-less truncations to cleave ubiquitin chains, while catalytic activity towards a peptide substrate is not impaired. It is corroborated by newly added data showing that the UBZ/MIU/zUBD region does interact with K63-chains (Suppl. Figure S6d-e). The second requirement is demonstrated by the loss of chain cleavage in an alpha-2/3 deletion mutant, while a quintuple mutant within the alpha-2/3 loop has no detrimental effect.

Last gel in Fig 3F is confusing (where is Ub4 b and if it is not cleaved).

The three gels of Figure 3f form a common panel, which shows the degradation of polymeric K63-chains to smaller units and finally to mono-ubiquitin. The reaction products Ub, Ub2, Ub3 and Ub4 form discernible bands, whose positions are indicated at the right hand side of the panel – even if these bands are not visible in the last gel of the panel. We have revised the figure legend to indicate this fact.

Also, it would be interesting to check the activities of the truncated mutants against diUB-K63 substrate, I would expect cleavage only with the presence of α 2-3 and ZUBD (each domain for the proximal and distal ubiquitin). This would confirm the role of α 2-3 in the K63 preference, and is a major point in the manuscript.

We have generated a TcZUP1 construct lacking the α 2/3 and zUBD domains (TcZUP³²⁵⁻⁵⁹²) and performed the requested experiment; the result is shown in Suppl. Figure S3d. As expected, the TcZUP³²⁵⁻⁵⁹² construct was inactive against K63-ubiquitin, while TcZUP²⁵⁷⁻⁵⁹², which does contain the α 2/3 and zUBD domains, is still active. This result shows the importance of the α 2/3 and zUBD domains for K63 specificity and is corroborated by Figure 3F, which shows that the TcZUP²⁸⁶⁻⁵⁹² construct (which lacks the zUBD but still contains α 2/3) is also inactive against K63 chains. In addition, the newly added pulldown results requested by reviewer 1 show that the α 2/3 - zUBD region does bind K63 chains (Suppl. Figure 6d,e).

Minor points: Are the proteins well purified ? There are many extra bands in the SDS-PAGE of figure 2.

Admittedly, the gels in figure 2 look confusing at first sight with several extra bands. Nevertheless, the purity of the enzyme preparations used in this manuscript is generally good, with the possible exception of full-length TcZUP. This protein has extensive unstructured regions in the N-terminal portion, which are targeted by bacterial proteases during expression - even when adding protease

inhibitors during purification. The only gels, where the resulting fragments are visible is figure 2f and 2h, all the other experiments with TcZUP were done using N-terminal truncations, which did not show this problem.

The additional bands in AtZUP preparations (figure 2d, 2e) are also due to cleavage by a bacterial protease, but in this case, cleavage is at a single well-defined site that we analyzed in detail (page 10, Suppl. Fig. S5) because we initially assumed it to be auto-cleavage. This processing event generates two bands (58kDa and 22 kDa in figure 2d,e) besides the uncleaved band (75 kDa in figure 2d,e). We have added arrowheads that facilitate the identification of these bands. This cleavage is visible in all AtZUP preparations, the size of the bands depends on the boundaries of the construct.

Finally, there are several unlabeled bands visible in figure 2e and 2g, which are derived from the substrate ubiquitin chains and their cleavage products. Since oligo-Ub chains of various linkage types are known to migrate differently in SDS-PAGE (each lane contains a different linkage), there is no easy way to label those bands.

The authors should consider a better labeling of the gels in Fig 2.

In the initial submission, we had inadvertently included a preliminary version of figure 2. We have now replaced it by the correct version, which should no longer have this problem.

The color of protein domains in the structure cartoons in Fig 3 should match the domain scheme.

We have changed the coloring of the domain scheme to match the colors in the structural cartoon.

The authors checked next the composition of the oxyanion hole, and say that a serine in that position enhances the catalytic activity of hZUP1 over glycine; and they also check the role of a tryptophan as a gatekeeper in the ubiquitin tail recognition. Although informative, it does not add anything to the substrate preference analysis.

Strictly speaking, this is correct. We nevertheless think that these results are noteworthy. Following the request of reviewer 1, we have now expanded the analysis of the oxyanion hole stabilization, including the determination of kinetic parameters for human TcZUP, human ZUP1, and the serine mutant. We think that this comparison helps to clarify the contributions by the active site (as opposed to the UBDs) to overall activity.

Minor points: Again SDS-PAGE gels should be better labeled.

The reaction of ZUP1 with Ub-PA is quite poor in figure 5h. I would expect a better yield.

We generally avoid using the absolute Ub-PA reactivity as a quantitative readout. We know from previous experiments that the quality of our Ub-PA preparations varies: they often contain a substantial portion of non-reactive material, leading to apparently incomplete reactions even with highly active enzymes. In most publications, Ub-PA reactivity is solely used qualitatively, showing only the end-point results after long-term incubation. In our figure 5h, we used the same Ub-PA batch and intentionally chose a shorter incubation time (1h) to allow a comparison between different enzyme activities rather than showing the end-points. We strongly recommend to only consider the difference in adduct formation, not the absolute quantities.

The N-terminal Zn finger in plants is essential for K48 cleavage, in contrast to Mug105 which has K48 preference without any extra domain (the gel in Fig6a is ugly and should be redone).

Figure 6a shows the degradation of a mixture of long ubiquitin chains (Ub_{6+}) by AtZUP. It indeed shows a large number of bands and might look confusing at first sight. However, all of the bands can be accounted for, and most of them are derived from the heterogeneous poly-ubiquitin substrate. In addition, as mentioned above (and documented extensively in the manuscript), the plant protease contributes three bands (one uncleaved and two cleavage products). To help the reader understand this complex gel, we have added arrowheads to indicate the protease-derived bands and have also indicated the range of the poly-Ub bands.

Finally, the role of the $\alpha 2/3$ is analyzed in ZUP1, which might be the essential interface for K63 chain preference of the ZUFSP members. However, the results with the quimeric Mug105 with ZUP1 N-terminal domain elements are discouraging, and a change in the specificity is not observed. All the results confirm the UB chain substrate preferences for the different ZUFSP members, but do not shed light to the structural mechanisms to reach chain preferences for each particular member. Only the $\alpha 2/3$ domain seems necessary for K63 cleavage. It would be really interesting a complex structure with a diUB-K63 substrate. Also, I would expect a better structural-based explanation for the Ub linkage preference in the plain Mug105 domain. What elements in the deubiquitinase domain are involved in chain preference ?.

We agree that a complex structure with a K63-diUb substrate would have been helpful and we have tried very hard to get one. Unfortunately, this complex did never form crystals in our hands, which might be due to the fact that the S1'-ubiquitin might not have a well-defined binding site. In the revised version, we have extended the discussion of the factors contributing to K63 binding.

REVIEWERS' COMMENTS

Reviewer #1 (Remarks to the Author):

In this revised version, previous concerns raised by this reviewer have been addressed adequately. The authors have made sincere efforts in addressing all the points raised in the previous round of review to the best of their ability. As a result, the manuscript appears to be much improved.

Minor points:

I would urge the authors to go through the current version carefully and weed out remaining typos (There may be just a few still lingering in this version. For example, in line 187, and in lines 227 and 230, 'conformation').

In line 266, which two chains 'still form the proper 3D structure'? Do the chains refer to the fragments produced upon proteolysis after Gly254? Please make it clearer.

In line 392, it may help to remind the readers that the G-to-S mutant does not show any appreciable effect on physiologically relevant polyubiquitin chain substrates.

Reviewer #3 (Remarks to the Author):

The authors have properly addressed all my initial concerns. I have no further questions about the manuscript.

Reviewer #1 (Remarks to the Author):

In this revised version, previous concerns raised by this reviewer have been addressed adequately. The authors have made sincere efforts in addressing all the points raised in the previous round of review to the best of their ability. As a result, the manuscript appears to be much improved.

Minor points:

I would urge the authors to go through the current version carefully and weed out remaining typos (There may be just a few still lingering in this version. For example, in line 187, and in lines 227 and 230, 'conformation').

We have carefully checked the manuscript and corrected the typos mentioned by the reviewer, as well as a few additional ones.

In line 266, which two chains 'still form the proper 3D structure'? Do the chains refer to the fragments produced upon proteolysis after Gly254? Please make it clearer.

We have rephrased this sentence to make it clear that we meant the two cleavage products.

In line 392, it may help to remind the readers that the G-to-S mutant does not show any appreciable effect on physiologically relevant polyubiquitin chain substrates.

We have added this information to the discussion section, line 392.

Reviewer #3 (Remarks to the Author):

The authors have properly addressed all my initial concerns. I have no further questions about the manuscript.

We thank the reviewers for their assessment